

# Carbon monoxide column retrieval for clear-sky and cloudy atmospheres: a full-mission data set from SCIAMACHY 2.3 µm reflectance measurements

Tobias Borsdorff[1], Joost aan de Brugh[1], Haili Hu[1], Philippe Nédélec[2], Ilse Aben[1], and Jochen Landgraf[1]

[1]SRON Netherlands Institute for Space Research, Utrecht, the Netherlands
[2]Laboratoire d'aérologie (LA), CNRS UMR-5560 et Observatoire Midi-Pyrénées, Université Paul-Sabatier,
Toulouse, France

*Correspondence to:* T. Borsdorff (t.borsdorff@sron.nl)

**Abstract.** We discuss the retrieval of carbon monoxide (CO) vertical column densities from clear-sky and cloud contaminated 2311-2338 nm reflectance spectra measured by the Scanning Imaging Absorption Spectrometer for Atmospheric Chartography (SCIAMACHY) from January 2003 until the end of the mission in April 2012. These data was processed with the Shortwave Infrared CO Retrieval algorithm SICOR that we developed for the operational data processing of the Tropospheric Moni-

toring Instrument (TROPOMI) that will be launched on ESA's Sentinel-5 Precursor (S5P) mission. This study complements previous work that was limited to clear-sky observations over land. Over the oceans, CO is estimated from cloudy-sky measurements only, which is an important addition to the SCIAMACHY clear-sky CO data set as shown by NDACC and TCCON measurements at coastal sites. For Ny-Ålesund, Lauder, Mauna Loa, and Reunion, a validation of SCIAMACHY clear-sky retrievals is not meaningful because of the high retrieval noise and the few collocations at these sites. This improves significantly

when considering cloudy-sky observations, where we find a low mean bias $\bar{b} = \pm 6.0$ ppb and a strong correlation between the validation data set and the SCIAMACHY data sets with a mean Pearson correlation coefficient $r = 0.7$. Also for land observations, cloudy-sky CO retrievals present an interesting complement to the clear-sky data set, which is less sensitive to the spatial representativeness of the satellite and validation measurement. For example, at the cities Teheran and Beijing the agreement of SCIAMACHY clear-sky CO observations with MOZAIC/IAGOS airborne measurements is poor with a mean

bias of $\bar{b} = 171.2$ ppb and $57.9$ ppb because of local CO pollution, which cannot be captured by SCIAMACHY. The validation improves significantly for cloudy-sky retrievals with $\bar{b} = 52.3$ ppb and $5.0$ ppb, respectively. This is due to a reduced retrieval sensitivity to CO below the cloud and so to the altitude range, which is mostly affected by strong local surface emissions. At the less urbanised region around the airport Windhoek, local CO pollution is less prominent and so MOZAIC/IAGOS measurements agree well with SCIAMACHY clear-sky retrievals with a mean bias of $\bar{b} = 15.5$ ppb, but can be even further improved

considering cloudy SCIAMACHY observations with a mean CO bias of $\bar{b} = 0.2$ ppb. Overall the cloudy-sky CO retrievals from SCIAMACHY short wave infrared measurements present a valuable addition to the clear-sky only data set. Moreover, the study represents the first application of the S5P algorithm for operational CO data processing on cloudy observations prior to the launch of the S5P mission.



## 1 Introduction

The Tropospheric Monitoring Instrument (TROPOMI) will be launched on board of the Copernicus Sentinel-5 Precursor (S5P) satellite. Besides the ultraviolet, visible and near infrared spectral range, it will measure Earth radiance and solar irradiances in the $2.3\,\mu m$ short wave infrared (SWIR) spectral range during its expected lifetime of seven years (Veefkind et al., 2012). The

TROPOMI SWIR spectrometer builds upon the Scanning Imaging Absorption Spectrometer for Atmospheric Chartography (SCIAMACHY) that was operational from January 2003 to April 2012 on ESA's ENVISAT satellite (Bovensmann et al., 1999). The SWIR measurements of TROPOMI and SCIAMACHY will be similar with respect to spectral coverage and resolution, but TROPOMI will have better spatial resolution ($7{\times}7\,km^2$), a larger signal-to-noise ratio, and daily global coverage. Therefore TROPOMI will extend the record of SWIR measurements from space, and because of its similarity to SCIAMACHY, it is

possible to use the same CO retrieval algorithm for both instruments and thereby provide a consistent data product for long-term study of CO (Borsdorff et al., 2016; Landgraf et al., 2016b).

The SWIR measurements around 2.3 μm of SCIAMACHY have been used to retrieve the vertical column densities of various atmospheric trace gases: HDO (Frankenberg et al., 2009; Scheepmaker et al., 2015), water vapor (Schrijver et al., 2009), and CO (Frankenberg et al., 2005; Buchwitz et al., 2004; Gloudemans et al., 2008; Gimeno García et al., 2011; Borsdorff et al.,

2016). In particular, the SCIAMACHY CO product has been used to analyse biomass burning events (Buchwitz et al., 2004), monitor the transport of atmospheric pollutants (Gloudemans et al., 2006), and detect pollution from mega cities (Buchwitz et al., 2007).

The major limitation of the SCIAMACHY CO data set is its high retrieval noise, which can exceed 100 percent of the retrieved value for individual columns. Hence, in practice the data needs to be averaged spatially and temporally to reduce

the noise contribution (de Laat et al., 2007; Gloudemans et al., 2006). The SCIAMACHY instrument has had issues with degradation and calibration in the SWIR spectral range, and efforts to retrieve CO are hampered by the failure of a significant number of detector pixels and the growth of a layer of ice on the detector array (Gloudemans et al., 2005, 2008). Borsdorff et al. (2016) discussed a re-calibration of the measurements and presented a full mission CO data set, which is restricted to clear-sky scenes over land. This SCIAMACHY CO data product for clear skies was validated with ground-based Fourier Transform

Spectrometer (FTS) measurements that were provided by the Network for the Detection of Atmospheric Composition Change (NDACC) and the Total Carbon Column Observing Network (TCCON) (de Laat et al., 2010; Borsdorff et al., 2016), and it was also validated with airborne measurements provided by the MOZAIC/IAGOS project (de Laat et al., 2012; Borsdorff et al., 2016). The successful validation of this recalibrated data set showed that SCIAMACHY observations reveal meaningful information about atmospheric CO over the full mission period.

SCIAMACHY's SWIR measurements of clear skies over land have a good sensitivity for the vertical column density of CO. However, the clear-sky CO data product requires a strict cloud filter over land and the wholesale rejection of all observations over oceans – due to the low reflectivity of the ocean surface in the SWIR. Hence, a retrieval that works for cloudy scenes is necessary in order to extend spatial coverage beyond the small fraction of SCIAMACHY observations which are made of clear skies over land.





Gloudemans et al. (2009) and Buchwitz et al. (2006) demonstrated that cloud contaminated measurements can be useful for CO retrievals, because the high reflectivity of clouds decreases retrieval noise. By using SCIAMACHY observations under cloudy conditions, Gloudemans et al. (2009) investigated the outflow of CO from Asia and Indonesia over the oceans. However, to properly interpret these measurements, one has to account for the shielding and scattering effect of clouds. For this

purpose, Vidot et al. (2012) and Landgraf et al. (2016b) proposed the Shortwave Infrared CO retrieval (SICOR) algorithm for the operational processing of CO data for the S5P mission. In addition to trace gas absorption, SICOR estimates two cloud parameters: cloud optical thickness and cloud height. These parameters are used in a two stream radiative transfer solver called 2S-LINTRAN (Landgraf et al., 2016b) to account for the changes to light path that occur due to scattering by clouds and aerosols. SICOR produces a data product that includes the retrieved CO column, its noise estimate and the column averaging

kernel. The averaging kernel reflects the vertical sensitivity of the retrieval to CO, and it depends on the cloud parameters that were estimated for the observed scene.

In this study, we apply the SICOR algorithm to measurements from SCIAMACHY's $2.3\,\mu\text{m}$ SWIR channel. Doing this allows us to evaluate the SICOR algorithm's performance on real cloud-contaminated measurements and to expand upon the data set produced by Borsdorff et al. (2016). Whereas that CO data set was limited to clear-sky retrievals over land,

here we provide SCIAMACHY CO data for cloudy scenes over the oceans and both clear and cloudy scenes over land. The new data product includes the column averaging kernels for individual measurements, and we use these to help validate the data product with vertical concentration profiles of CO that were measured by the MOZAIC/IAGOS project. We also validate the data product with TCCON and NDACC ground-based FTS measurements. The paper is structured as follows. Section 2.1 introduces the inversion of vertical column densities by profile scaling, Sect. 2.2 explains the retrieval setup for

the SCIAMACHY measurements, and Sect. 2.3 analyses the quality of the retrieved cloud parameters. In Sect. 3.1 we present the data validation using MOZAIC/IAGOS aircraft measurements while Sect. 3.2 focuses on the validation with NDACC and TCCON measurements. A summary and conclusions are given in Section 4.

## 2 Inversion method

The inversion of CO vertical column densities from SCIAMACHY's $2.3\,\mu\text{m}$ reflectance spectra utilises the profile scaling

approach. This approach has been applied to SCIAMACHY data before by Gloudemans et al. (2008) and Borsdorff et al. (2016), and its regularisation type is discussed in detail by Borsdorff et al. (2014). Here, we summarise the aspects of inversion theory that are needed for the later discussion.

### 2.1 Profile scaling approach

The inversion is based on the assumption that the $m$-dimensional measurement $\boldsymbol{y}_{\text{meas}}$ can be described by the forward model

$\boldsymbol{F}$ in the bounds of the measurement error $\boldsymbol{e}_y$, namely

$$\boldsymbol{y}_{\text{meas}} = \boldsymbol{F}(\boldsymbol{x}, \boldsymbol{b}) + \boldsymbol{e}_y. \qquad (1)$$



Here, the vector $b$ comprises forward model parameters, which are known a priori. The state vector $x$ contains all parameters to be retrieved including the column density of CO and other trace gases.

The forward model $F$ requires as input the vertical concentration profile of the atmospheric trace gases, which we obtain by scaling a reference profile $\rho_{ref}$ with the corresponding column density of the trace gas in $x$. The reference profile does not change during the retrieval and is normalised by its vertical column density $c$,

$$c = C\rho_{ref} , \tag{2}$$

where the $n$-dimensional vector $C = (1,\ldots,1)^T$ approximates the vertical integration assuming that the entries of the CO profiles are given in subcolumns.

Equation (1) is inverted with respect to state vector $x$ using the least squares fitting approach, where we apply the Gauss-Newton algorithm to account for the non-linearity of the forward model. In the linear approximation the solution $x_{ret}$ can be expressed by the gain matrix $\mathbf{G}$ of the inversion,

$$x_{ret} = \mathbf{G}y_{meas} . \tag{3}$$

The retrieved vertical column density $c_{ret}$ is an element of the solution vector $x_{ret}$ and describes an effective column density due to the regularisation inherent to the profile scaling approach. In the linear approximation, the effective column and the true atmospheric abundance are related by the equation,

$$c_{ret} = \mathbf{a}_{col}\rho_{true} + e_c, \tag{4}$$

where $\mathbf{a}_{col}$ is the column averaging kernel, $e_c$ is the column retrieval error due to the measurement error $e_y$ and $\rho_{true}$ is the true trace gas profile. Borsdorff et al. (2014) presented a numerically efficient algorithm for calculating $\mathbf{a}_{col}$. The column averaging kernel represents an weighted integration of the true vertical profile taking into account the particular retrieval sensitivity. The differences between the true column, $c_{true} = C\rho_{true}$, and the effective column, $c_{eff} = \mathbf{a}_{col}\rho_{true}$, cannot be inferred from the measurement and is also known as the null-space or smoothing error of the retrieval (Borsdorff et al., 2014; Rodgers, 2000),

$$e_{null} = (C - \mathbf{a}_{col})\rho_{true}. \tag{5}$$

Finally, the measurement noise described by the measurement covariance matrix $\mathbf{S}_y$ introduces noise on our retrieval product, which is characterised by the retrieval noise covariance matrix,

$$\mathbf{S}_x = \mathbf{G}\mathbf{S}_y\mathbf{G}^T. \tag{6}$$

In this manner, we have defined all diagnostic tools for our retrieval. A detailed overview of the profile-scaling approach is given in Borsdorff et al. (2014).

## 2.2 Retrieval Settings

For SCIAMACHY CO retrievals, the SICOR algorithm settings are very similar to those of the TROPOMI CO data processor (Landgraf et al., 2016b, a) but with adaptions to account for the instrument degradation of the SCIAMACHY instrument





(Borsdorff et al., 2016). It comprises two main processing steps: first a non-scattering retrieval for cloud detection and subsequently a physics-based scattering retrieval to infer CO column abundances. To account for the limited radiometric accuracy and precision of SCIAMACHY measurements, we choose the spectral window 2311-2338 nm (Borsdorff et al., 2016) for both retrieval steps. This represents a wider retrieval window compared to the TROPOMI settings, which are 2315-2324 nm for the non-scattering retrieval and 2324-2338 nm for the physics-based retrieval (Landgraf et al., 2016b).

The settings for the non-scattering retrieval are identical with the one described by Borsdorff et al. (2016) and in first instance we retrieve the column density of CO, $CH_4$, $H_2O$, and HDO without accounting for atmospheric scattering and clouds in particular. The difference between the retrieved $CH_4$ column and the a priori $CH_4$ column from the chemical transport model TM5 (Williams et al., 2013, 2014) reflects the light-path effect of clouds on the observations due to shielding and photon path enhancement by multi-scattering (Vidot et al., 2012). Moreover for the subsequent retrieval of CO, we use the retrieved non-scattering $CH_4$ column to provide a first estimate for the cloud height. For this purpose we integrate the TM5 vertical $CH_4$ profile from the cloud center to the top of the model atmosphere and adjust the cloud height to match the retrieved non-scattering methane column.

The subsequent physics-based retrieval is described in detail by Landgraf et al. (2016b). It estimates the trace gas columns of CO, $H_2O$, and its isotopologue HDO simultaneously with the cloud height $h_{cld}$ and cloud optical thickness $\tau_{cld}$ assuming a fully overcast scene. To estimate the cloud parameters from the SWIR measurements, the atmospheric $CH_4$ vertical profile concentration is fixed to accurate model estimates provided by the TM5 model. The physics-based retrieval simulates a cloud as a horizontally homogeneous scattering layer with a triangular height profile in optical depth with a fixed half width of 1.5 km and infers the optical thickness and the cloud height from the methane absorption signal. The remaining a priori information and the molecular absorption spectroscopy is described by Borsdorff et al. (2016).

## 2.3 Evaluation of the retrieved cloud parameters

Compared to the previous work by Borsdorff et al. (2016), the new aspect of this study is the retrieval of CO column densities from cloudy SCIAMACHY measurements, where we infer cloud parameters from the 2.3 μm methane absorption band. The effect of clouds on the retrieved CO column density is described with the column averaging kernels for each individual measurement. Obviously the retrieved cloud properties are effective parameters and so do not necessarily represent the real atmospheric situation but describe clouds, cirrus and aerosols such that the simulated lightpath is sufficiently accurate to retrieve CO within the required accuracy. In that sense the retrieved cloud parameters depend on the forward model of the retrieval algorithm and the spectral fit window, which makes their verification difficult.

To gain first experience with the SICOR cloud parameters, we compared the retrieved cloud height $h_{cld}$ with the SCIAMACHY cloud top height product, processed with the Fast Retrieval Scheme for Clouds from the Oxygen A Band (FRESCO) algorithm. In contrast to SICOR, this algorithm infers cloud top height and cloud fraction from the SCIAMACHY the Oxygen A band around 760 nm assuming an elevated Lambertian reflector as a cloud model (Wang et al., 2008). This comparison is particular worthwhile since SCIAMACHY's NIR measurements do not suffer from the formation of an ice layer on the detector module like the SWIR measurements (Gloudemans et al., 2008). Figure 1 shows a scatter plot of the SICOR and FRESCO





cloud heights for one year of SCIAMACHY observations in year 2003 for an ocean region with a latitude range from $30°$ S to $0°$ and a longitude region from $30°$ W to $0°$. We only consider measurements with a significant cloudiness indicated by SICOR cloud optical depth of $\tau_{cld} > 1$. The cloud parameters are highly correlated with a Pearson correlation coefficient $r = 0.9$. A linear regression provides a slope of $1.01$ and an intercept of $300\,\mathrm{m}$, indicating a valuable cloud product from the SWIR

measurements of SCIAMACHY. However, this comparison should not hide the fact that the validity of the SICOR effective cloud parameters can only be demonstrated by comparing the SCIAMACHY CO data product for cloudy observations with independent validation measurements, which is the subject of the next section.

In the following, we use the SICOR cloud parameters to classify the retrievals with respect to cloudiness. We distinguish between three CO retrieval conditions: (1) clear-sky observations over land with $\tau_{cld} < 0.3$ and $h_{cld} < 1$ km, (2) cloudy observa-

tions contaminated by optically thick low clouds with $\tau_{cld} > 2$ and $h_{cld} < 1.5$ km, and (3) cloudy observations contaminated by optically thick high clouds with $\tau_{cld} > 2$ and $1.5$ km $< h_{cld} < 8$ km. For these categories, Fig. 2 shows typical column averaging kernels selected from the SCIAMACHY CO retrievals over Australia for the year 2003. Column averaging kernel values of one represent the ideal value for a vertical integration (see Eq. 2) and values $< 1$ indicate reduced retrieval sensitivity e.g. because of atmospheric shielding by clouds and aerosols. Values $> 1$ are typical for a profile scaling approach. In case of an optically

thick cloud the profile is adjusted at all altitudes including CO below the cloud using only the measurement sensitivity above the cloud. This means that the sensitivity of the retrieved CO column with respect to the CO concentration above the cloud is enhanced $(> 1)$ and at the same time the sensitivity for CO below the cloud is reduced $(< 1)$. These features are clearly reflected in Fig. 2.

So under clear-sky conditions we can retrieve CO vertical column densities with a sensitivity close to 1 at all altitudes (see

yellow line in Fig. 2). Hence, the corresponding null-space error is small (Buchwitz et al., 2004; Gloudemans et al., 2008) and a direct comparison of the retrieved column with CO column reference measurements is possible while neglecting the effect of column averaging kernels (Borsdorff et al., 2016). For cloudy retrievals the comparison must account for averaging kernel effects. Clouds can easily lead to null-space errors $> 30$ % depending on the discrepancy between true vertical trace gas profile and the reference profile to be scaled by the retrieval (Borsdorff et al., 2014).

Generally, the null-space error does not limit the data use if the averaging kernel is applied properly. For example, for data assimilation the averaging kernel provides the altitude sensitivity, which is needed to adequately adjust the atmospheric state. Also for the validation of the CO data product, the null-space error does not impose principle limitation. Using Eq. 4, we can account for cloud effects in our retrieval when comparing satellite measurements with independent profile soundings. Wassmann et al. (2015) demonstrated this strategy for ozone column retrievals from GOME-2 measurements using ozonesonde

measurements for validation. In practice, this approach is often hampered by the small number of available validation measurements.

CO retrievals for cloudy conditions represent an interesting addition of our data product because of the different vertical sensitivity, the retrieved CO column probes different altitude ranges and so providing complementary information. Additionally, the radiometric precision of SWIR measurements for cloudy scenes is generally much better than for clear-sky observations

because of the brightness of clouds. In the $2.3\,\mu\mathrm{m}$ spectral range this represents an important asset for data exploitation, partic-





ularly for SCIAMACHY, which suffers from a poor radiometric precision. Also for TROPOMI, with a much better radiometric performance, CO retrievals over cloudy scenes represent an important addition for ocean scenes where due to the low reflectivity of ocean water no information about atmospheric CO can be inferred from clear-sky observations.

To illustrate this, Fig. 3 shows the signal-to-noise ratio (SNR) and CO retrieval noise as a function of the SICOR cloud optical thickness $\tau_{cld}$ for an ocean region with a latitude range from $30°$ S to $0°$ and a longitude range from $30°$ W to $0°$. For a cloudless scene "$\tau_{cld} \to 0$", the signal-to-noise is less than 25 and the CO retrieval error is very high. Already for SICOR cloud optical thickness of 1-2 the SNR rises above 100 and the CO retrieval noise reduces to $5 \cdot 10^{17}$ molec cm$^{-2}$. For higher cloud optical thickness, the SNR and CO precision saturate around $\tau_{cld} = 8$ where the SNR reaches 250 and the corresponding CO retrieval noise is about $2 \cdot 10^{17}$ molec cm$^{-2}$. Hence for SCIAMACHY, our CO retrieval algorithm for cloudy conditions permits the derivation of CO over the ocean, and in addition has the potential to improve the CO product over land.

## 3 Validation

The quality of our CO retrieval for cloudy atmospheres needs to be demonstrated through validation. Here we begin by validating the SCIAMACHY CO retrievals with airborne profile measurements of the MOZAIC/IAGOS project, based on in-situ measurements on commercial airliners (Nédélec et al., 2003). Following this we complete our validation using CO ground-based measurements of the TCCON and NDACC networks. To ensure the quality of the SCIAMACHY CO column retrievals, we perform an a posteriori quality filter on the individual CO retrievals:

1. The $\chi^2$ values of the spectral fit must be $< 40$.

2. The mean signal-to-noise ratio of the measurements in the fit window must be $> 20$.

3. The noise $\epsilon$ of the retrieved CO, HDO and H$_2$O column must be below an upper threshold, namely $\epsilon_{CO} < 1 \times 10^{19}$ molec cm$^{-2}$, $\epsilon_{HDO} < 1 \times 10^{20}$ molec cm$^{-2}$ , and $\epsilon_{H_2O} < 4 \times 10^{22}$ molec cm$^{-2}$.

### 3.1 Airborne CO profile measurements (MOZAIC/IAGOS)

The MOZAIC/IAGOS project provides profile measurements of reactive gases performed on board of long-distance passenger airliners. Since 1994, in-situ profile measurements are performed during the ascent and descent phases of more than 40,000 flights. These observations are used to derive vertical column densities of CO with a precision of about 5 % (Nédélec et al., 2003). The recorded profiles do not represent a strict vertical intersection of the atmosphere but at lower altitudes atmospheric constituents are measured close to the airport where at higher altitudes the distance to the airport may reach 200-400 km. The study by de Laat et al. (2014) showed that in most cases the corresponding collocation errors with SCIAMACHY observations imposed by the aircraft ascent and descent phases average out to a large extend when analysing temporal averages.

In this study, we consider the vertical profile measurements at three cities: Teheran and Beijing, which are known to be affected by strong local pollution events (de Laat et al., 2012), and Windhoek, which is situated in an area with higher surface albedo and thus high radiometric precision of SCIAMACHY clear-sky measurements. Moreover, at Windhoek the atmospheric



CO abundance is only slightly influenced by local pollution resulting in spatially homogeneous CO fields. Table 1 provides more details on these sites. From the MOZAIC/IAGOS data set, we select only those profiles that cover at least an altitude range up to $9\,km$ with data gaps $< 1\,km$. This follows the procedure of de Laat et al. (2014) and ensures the availability of sufficient CO profile information to validate our data product. Above the maximum flight altitude, the CO profiles are extended

by simulated CO profiles of the TM5 chemical transport model. Furthermore, towards the ground level we extend the profiles assuming a constant CO mixing ratio. When the CO column density is derived from the aircraft measurement in this manner its accuracy is estimated to be a few percent (de Laat et al., 2012).

To collocate SCIAMACHY retrievals with MOZAIC/IAGOS measurements, we select all SCIAMACHY CO retrievals with a ground pixel in a radius of 850 km around the airport site. The temporal collocation criterium is chosen dynamically for each

individual MOZAIC/IAGOS measurement. Centered around the recording time of the aircraft measurements, the temporal collocation window is chosen such that the average of all spatially collocated SCIAMACHY measurements have a precision $< 10^{17}$ molec cm$^{-2}$. Therewith, we ensure that the precision of the SCIAMACHY validation measurements is sufficient for our purpose. We consider SCIAMACHY measurements for clear-sky and cloudy conditions with high, optically thick clouds corresponding to condition (1) and (3) defined in Sec.2.3 and calculate the collocated MOZAIC/IAGOS CO column

density from the corresponding profile in two ways: first we perform the vertical integration of the MOZAIC/IAGOS profile after accounting for elevation differences between the SCIAMACHY ground scene and the airport site. Second, we apply the column averaging kernels of all collocated SCIAMACHY observations to the MOZAIC/IAGOS profile and calculate their average. Finally, we average the collocated SCIAMACHY CO column densities accordingly. Figures 4 and 5 show these time series for the three airport sites. Per site, we calculate the mean and standard deviation of the CO time series as a diagnostic

tool, which is shown on Fig. 6 for the clear-sky and cloudy SCIAMACHY data sets. The figure clearly shows that applying the SCIAMACHY column averaging kernels to the MOZAIC/IAGOS profiles improves the validation for all sites for both clear-sky and cloudy-sky conditions. For clear-sky measurements, this bias improvement is small and is about 9.5 ppb for Teheran, 4.0 ppb for Beijing, and only 0.5 ppb for Windhoek, which confirms the findings by Borsdorff et al. (2014); Gloudemans et al. (2005); Buchwitz et al. (2004). For cloudy-sky measurements, we notice a significant improvement of the bias, 122.2 ppb for

Teheran , 22.6 ppb for Beijing, and 7.3 ppb for Windhoek.

The time series in Fig. 4 show strong outliers in the MOZAIC/IAGOS CO column densities, which are not found by SCIA-MACHY. de Laat et al. (2012) explained this with strong local pollution at the airports, which affects the airborne measurements but are not seen by the satellite due to the coarse spatial resolution. Figure 6 confirms this reasoning. Applying the column averaging kernels of the cloudy-sky retrievals to MOZAIC/IAGOS measurements, the enhanced CO column values are strongly

reduced for both Teheran and Beijing and brings the aircraft and satellite observations into much better agreement. Thus confirms that the column averaging kernel of the cloudy-sky retrieval describes well the cloud shielding of the lower atmosphere (see Fig. 2) where the local pollution is primarily located.

Another interesting feature of the SCIAMACHY cloudy-sky retrievals is that the agreement between SCIAMACHY and MOZAIC/IAGOS measurements integrated by the column averaging kernel is better for Beijing than for Teheran. At the

same time the SCIAMACHY mean CO column mixing ratio at Teheran is about 43.0 ppb smaller than for Beijing with about




16.0 ppb less scatter. This may hint at a larger representation errors for Teheran than for Beijing, meaning that the aircraft and satellite observations represent different air masses due to their different spatial and temporal sampling and averaging. Beijing with 18.6 million urban inhabitants and an urban area of about $1400 \, \text{km}^2$ has about double the size of Teheran with an urban population of about 8.8 million and an urban area of $730 \, \text{km}^2$. Moreover the metropolitan area in Beijing is much larger than that of Teheran, indicating that the CO sources in Beijing may be spatially more extended than those in Teheran. This would mean that SCIAMACHY captures the enhanced CO concentrations over Beijing better than those over Teheran and would explain the differences we see in Fig. 6 for both sites.

We complete our SCIAMACHY CO validation that is based on MOZAIC/IAGOS aircraft measurements by analysing observations from the airport at Windhoek. Figure 5 shows good agreement between SCIAMACHY and MOZAIC/IAGOS measurements with a clear seasonality. We notice a small improvement when applying the SCIAMACHY cloudy-sky averaging kernels to the aircraft measurements, which removed some high values in the time series. The mean and standard deviation in Fig. 6 show that SCIAMACHY cloudy-sky and clear-sky retrievals agree well with the MOZAIC/IAGOS data for Windhoek.

At this site the reference profile $\rho_{\text{ref}}$ from the TM5 model is more accurate and hence applying the averaging kernels to the MOZAIC/IAGOS validation measurements is less important. Furthermore, the spatial CO distribution is less affected by strong local sources, which reduces representation errors of both data sets.

### 3.2 Ground-based Fourier Transform Spectrometers (NDACC/TCCON)

The TCCON and NDACC networks perform direct sunlight measurements with ground based Fourier transform spectrometers under clear-sky conditions. The Infrared Working Group (IRWG) performs measurements in the mid-infrared spectral range at $4.8 \, \mu\text{m}$ and is part of NDACC (http://www.ndsc.ncep.noaa.gov/). Thereby, it supplies CO vertical column densities that we transformed to column averaged mixing ratios by calculating the air column from the surface pressure at the station sites. The TCCON instruments measure in the same spectral range that SCIAMACHY does with the aim of deriving the vertical column densities of trace gases including CO to high precision (Wunch et al., 2010; Wunch et al., 2011).

Because of the lack of profile information, it is not possible to apply averaging kernels, and NDACC/TCCON data must be compared directly with the SCIAMACHY cloudy-sky retrievals. The comparison depends critically on the CO a priori profile information, and so we select near-coast measurement sites in remote areas, where the TM5 model prediction of the relative CO profile is most reliable Ny-Ålesund, Wollongong (Griffith et al., 2014), Lauder (Sherlock et al., 2014), Mauna Loa, and Reunion . More details about the stations are given in Table 2. Figures 7 and 8 show time series of 30-day median values derived from the ground-based FTS measurements and the collocated SCIAMACHY retrievals. Using the same criteria as in Sec. 3.1, missing values of the ground-based FTS time series were interpolated (open circles) using the TM5 model as described by Borsdorff et al. (2016).

The upper panel of Fig. 7 illustrates that data validation with ground-based measurements sites on small islands like Mauna Loa and Reunion are challenging for the clear-sky SCIAMACHY CO retrievals over land. Only a few satellite land observations can be collocated with the ground-based measurements. Moreover, these measurements have low surface albedo, which results in a large retrieval noise $> 100.0 \, \text{ppb}$ for single soundings. So for these locations, only a few 30-day median values can be





determined, which are still dominated by the retrieval noise as shown in the left panel of Fig. 9. The scatter of individual SCIAMACHY retrievals is described by the half difference of the $15.9^{th}$ and the $84.1^{th}$ percentile, which is written as $\epsilon_S$ and taken to be an analogue for the standard deviation of a normal distribution. Further, $\epsilon_N$ is the mean retrieval noise of the single measurements of the monthly medians.

The correlations between the FTS and clear sky SCIAMACHY measurements over land are poor (about $r = 0.3$) and the high standard error ($> 17.0\,\mathrm{ppb}$) indicates that under these conditions even the overall mean value is difficult to estimate. The situation improves for measurement sites at larger islands like Lauder and Ny-Ålesund. We found more collocations between SCIAMACHY land observations and FTS measurements (Fig. 8) which led to smaller standard error of the mean ($< 8.0\,\mathrm{ppb}$), and a more reliable estimation of the overall mean. However, the 30-day median values are still dominated by the retrieval

noise due to the low surface albedo of the observed scenes, and so the correlations between the FTS and SCIAMACHY measurements are still low (about $r = 0.3$, see Fig. 9). For SCIAMACHY clear-sky measurements over land, we achieved the best results for FTS measurements at Wollongong with high surface albedo. The upper panel of Fig. 8 shows that in this case the 30-day CO medians agree well with the FTS measurements. Moreover, the global mean can be estimated with a small standard error and the Pearson correlation coefficient for FTS and SCIAMACHY measurements is $r = 0.5$ indicating that the

CO seasonality of the FTS measurements can be captured well by the SCIAMACHY observations.

     When analysing SCIAMACHY cloudy-sky CO retrievals with $1.5\,\mathrm{km} < h_{cld} < 8\,\mathrm{km}$, $\tau_{\mathrm{cld}} > 2$ corresponding to condition (2) defined in Sec. 2.3, the comparison improves for all stations (see lower panel of Figs. 7 and 8). Due to the high reflectivity of clouds, the single measurement retrieval noise is reduced to $< 30.0\,\mathrm{ppb}$ and much more ground-pixels can be collocated with SCIAMACHY. Figure 9 shows that the standard error of the mean bias falls below $2.0\,\mathrm{ppb}$. For Lauder, the mean bias improves

from $\bar{b} = -12.0$ to $\bar{b} = -2.0\,\mathrm{ppb}$ but stays nearly the same for Ny-Ålesund. In both cases, the correlation increases to $r = 0.6$ and $0.7$ respectively. For Wollongong, the mean bias for clear-sky and cloudy-sky retrievals is comparable (about $4.0\,\mathrm{ppb}$), but the correlation improves from about $r = 0.5$ to $0.8$ for the cloudy retrievals. Considering all reported cloudy-sky retrievals, we estimate a mean bias $\bar{b} = -6.0\,\mathrm{ppb}$. This may be due to remaining effects of the retrieval noise or due to the loose collocation criteria. Also issues due to erroneous radiometric calibration of SCIAMACHY SWIR measurements cannot be excluded.

The presented validation analysis provided sufficient confidence to re-process the global full-mission SCIAMACHY CO data set including clear-sky and cloudy-sky observations. The full-mission averaged CO product is shown in Fig. 10 and indicates that the cloudy-sky ocean retrieval adds valuable information to the clear-sky data product. For example, over the Atlantic Ocean the CO outflow due to biomass burning in central Africa becomes clearly visible as well as the CO transport over the Pacific Ocean due to pollution from China and the wildfires in Indonesia. All this reveals information about the global

transport of pollution in the atmosphere. Future research must demonstrate the merit of this observations for studying global transport of pollution in the Earth atmosphere.



## 4 Conclusions

In this study, we derived a full mission SCIAMACHY CO column data set that comprises retrievals from clear-sky and cloudy-sky reflectance measurements in the spectral range 2311-2338 nm over land and ocean scenes. The inversion uses the SICOR CO retrieval code that is developed for the operational data processing of the Sentinel 5 precursor mission. It allows us to

retrieve effective cloud parameters simultaneously with trace gas columns. The data product includes the CO column and its column averaging kernel for each individual sounding, and so it provides information on the vertical CO retrieval sensitivity, which changes with the cloudiness of the observed scene. This study focused on the validation of SCIAMACHY CO retrievals for cloudy-sky observations with MOZAIC/IAGOS aircraft measurements and TCCON and NDACC ground based observations. It represents an extension to cloudy skies of the previous work by Borsdorff et al. (2016), presenting a similar data set

for clear-sky scenes over land.

The effective cloud parameters (cloud height $h_{cld}$, cloud optical thickness $\tau_{cld}$) are needed to properly simulate the atmospheric lightpath as part of the CO retrieval. Generally, the validation of effective cloud parameters is difficult, and in this study the cloud parameters are validated implicitly by the validation of the CO data product from cloudy-sky SCIAMACHY observations with independent atmospheric measurements. To gain confidence in the cloud product, we compared the cloud

height of the SICOR data product from one year of SCIAMACHY observations over the ocean with the corresponding cloud height of the FRESCO data product, which is inferred from SCIAMACHY measurements of the $O_2$ A band. We found a strong correlation ($r = 0.9$) between both products and an overall offset of the SICOR cloud height versus FRESCO of $300\,\mathrm{m}$. This agreement shows that the cloud height retrievals from the different spectral ranges of SCIAMACHY are consistent even though the SWIR measurements are affected by radiometric degradation of the instrument.

SICOR uses the profile scaling approach to infer CO vertical column densities from the measurement, which involves a regularisation of the inversion problem. When interpreting the retrieved column as an estimate of the truth, a null-space error is introduced. This null-space error depends on the accuracy of the profile shape to be scaled by the inversion. Generally for clear-sky observations the null-space error is small, but for cloudy conditions it can easily exceed 30 percent. Here, clouds shield the atmosphere below and the a priori CO profile shape is used to add the lacking information. The sensitivity of the retrieved

CO column with respect to the true CO density is provided by the column averaging kernel for each individual retrieval. For the validation, we can thus interpret the retrieved column as a vertically integrated CO column density weighted by the column averaging kernel and so the null-space error becomes less relevant for the comparison of the retrieval product with independent atmospheric measurements of the CO profile.

Validating SCIAMACHY retrievals with MOZAIC/IAGOS airborne measurements confirmed the approach. Direct com-

parison of the MOZAIC/IAGOS CO columns estimated at Beijing, Teheran, and Windhoek with collocated SCIAMACHY clear-sky CO retrievals of the CO column showed a small bias of $0.5 - 9.5\,\mathrm{ppb}$, in agreement with previous studies. However for cloudy SCIAMACHY observations, the bias exceeds $120.0\,\mathrm{ppb}$ for Teheran and $30.0\,\mathrm{ppb}$ for Beijing. This indicates the large relevance of the null-space error for the comparison of cloudy observations with other data. The difference became much smaller when we accounted for the retrieval sensitivity in the comparison by applying the column averaging kernels to the air-





borne measurements. Here the bias reduced to 50.0 ppb for Teheran and 8.0 ppb for Beijing. The remaining error for Teheran we attribute to the different sampling of different air masses by the satellite and aircraft measurements. For Beijing with spatially much more extended CO surface emission, these errors are less relevant. For Windhoek, the comparison of aircraft and satellite observations shows differences of 0.2 ppb when the averaging kernel is applied and 8.0 ppb otherwise. For this site,

the vertical shape of the a priori CO profile is well known and so explains the smaller bias for this site.

We completed our validation study using ground-based FTS measurements from the NDACC and TCCON networks for the coastal sites Ny-Ålesund, Lauder, Mauna Loa, Reunion and Wollongong. At these validation sites, independent measurement of the CO profile is not available, and so we assumed a certain shape of the CO profile. Hence, we compare directly the column densities from FTS with those of SCIAMACHY measurements for both clear-sky conditions and cloudy conditions

with low clouds. Considering only clear-sky SCIAMACHY observations over land at Lauder, Mauna Loa, and Reunion, the comparison was dominated by the SCIAMACHY retrieval noise because of the low surface albedo and the insufficient number of collocations. Filtering the SCIAMACHY data set to select for optically thick low clouds over land and oceans, we found good agreement with the FTS ground based measurements with a correlation coefficient of $r = 0.6 - 0.7$.

For Wollongong the high surface albedo led to CO retrievals from SCIAMACHY clear-sky measurements that compare

well with FTS ground based measurements with a mean bias of $\bar{b} = -4.3$ ppb and a correlation of $r = 0.5$. Even for this site, the cloudy retrievals improved the comparison with a high correlation of $r = 0.8$. The mean bias stayed the same within its uncertainty, which demonstrated the overall consistency of the clear-sky and cloudy-sky data product.

Finally we processed the full-mission data record of SCIAMACHY SWIR measurements. This data-set demonstrates clearly the asset of cloudy-sky CO retrievals over oceans providing a global coverage of the SCIAMACHY CO data product. It is the

first time that the operational TROPOMI CO algorithm is tested successfully on real data for clear-sky and cloudy atmospheres, which is an important milestone for the preparation of the S5P mission. Although TROPOMI SWIR measurements will have the same spectral coverage and resolution as SCIAMACHY, we expect a much better data product due to its better radiometric performance and its better spatial resolution and sampling. After launch of SP5 this techniques used in this study will allow us to use IAGOS/MOZAIC and FTS measurements by the TCCON and NDACC networks for the validation of TROPOMI

CO. Additionally, using the same retrieval approach for SCIAMACHY and TROPOMI CO retrievals will help to provide a consistent long-term CO data set for both missions.

## 5   Data availability

The full-mission SCIAMACHY CO data set of this study including clear-sky and cloudy-sky observations is available for download at ftp://ftp.sron.nl/pub/pub/DataProducts/SCIAMACHY_CO/. The underlying data of the figures presented in this

publication can be found at ftp://ftp.sron.nl/open-access-data/. TCCON data is accessible from http://tccon.ornl.gov/ and NDACC measurements from http://www.ndsc.ncep.noaa.gov/. MOZAIC/IAGOS profiles can be download from http://www.iagos.fr.





*Acknowledgements.* SCIAMACHY is a joint project of the German Space Agency DLR and the Dutch Space Agency NSO with contribution of the Belgian Space Agency. The work performed is (partly) financed by NSO through the SCIAvisie project and in collaboration with the SCIAMACHY Quality Working Group (SQWG) by ESA. We acknowledge the European Commission for the support to the MOZAIC project (1994–2003) and the preparatory phase of IAGOS (2005–2012), the partner institutions of the IAGOS Research Infrastructure (FZJ, DLR,

5  MPI, KIT in Germany, CNRS, CNES, Météo-France in France and the University of Manchester in UK), ETHER (CNES-CNRS/INSU) for hosting the database, the participating airlines (Lufthansa, Air France, Austrian, China Airlines, Iberia, Cathay Pacific) for the transport free of charge of the MOZAIC/IAGOS instrumentation. MACC data were obtained from http://www.copernicus-atmosphere.eu. We acknowledge the NDACC-IRWG and TCCON ground-based FTS networks for providing data. This work was carried out on the Dutch national e-infrastructure with the support of SURF Cooperative.



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

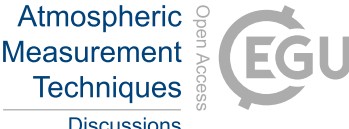



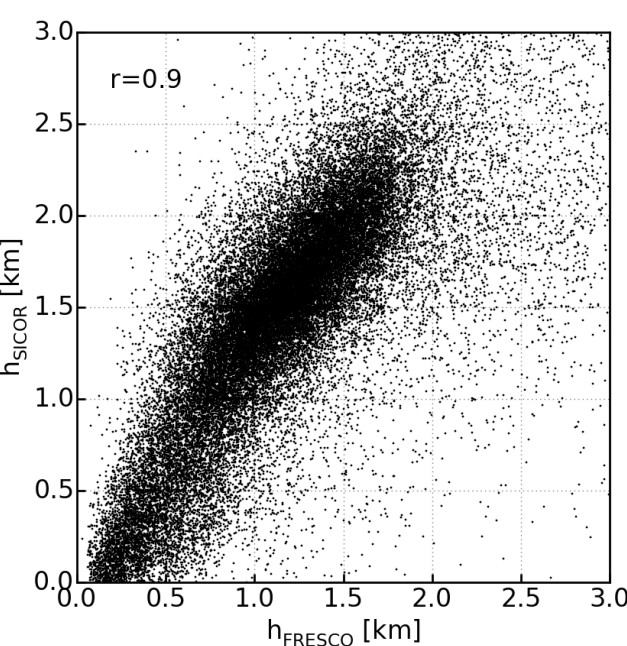

**Figure 1.** Scatter plot of the retrieved cloud height $h_{\mathrm{SICOR}}$ in km with the one provided by Fresco $h_{\mathrm{FRESCO}}$. For the year 2003 over the ocean in the latitude/longitude box [(30°S,30°W), (0°,0°)] with cloud optical thickness > 1 are shown.





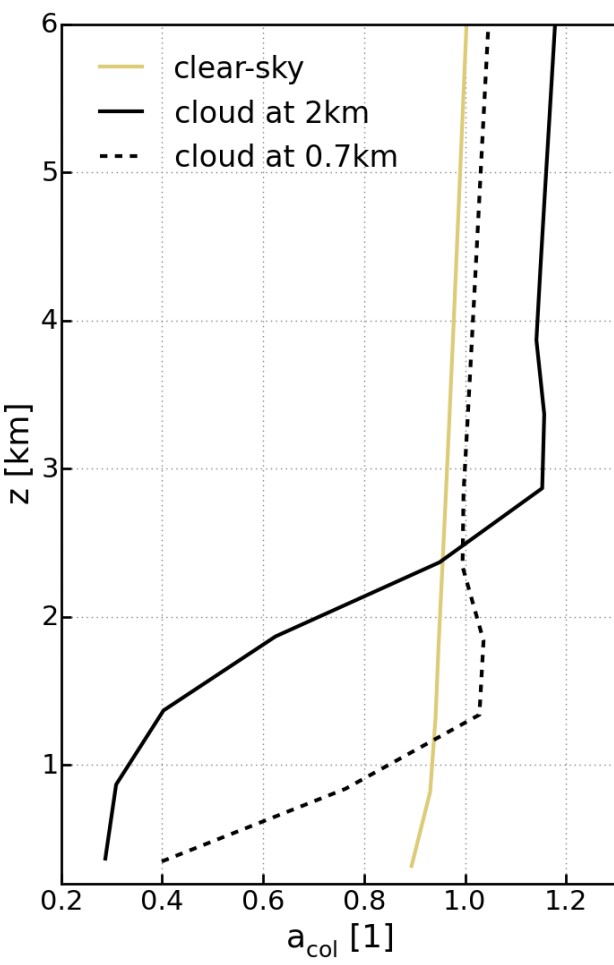

**Figure 2.** Example SCIAMACHY CO column averaging kernels over Australia in 2003 as a function of altitude for the following cases: clear-sky (yellow line), a cloud at 0.7 km (dashed line), and a cloud at 2 km (solid line). Cloud optical thickness of 2.5 for the relevant cases.





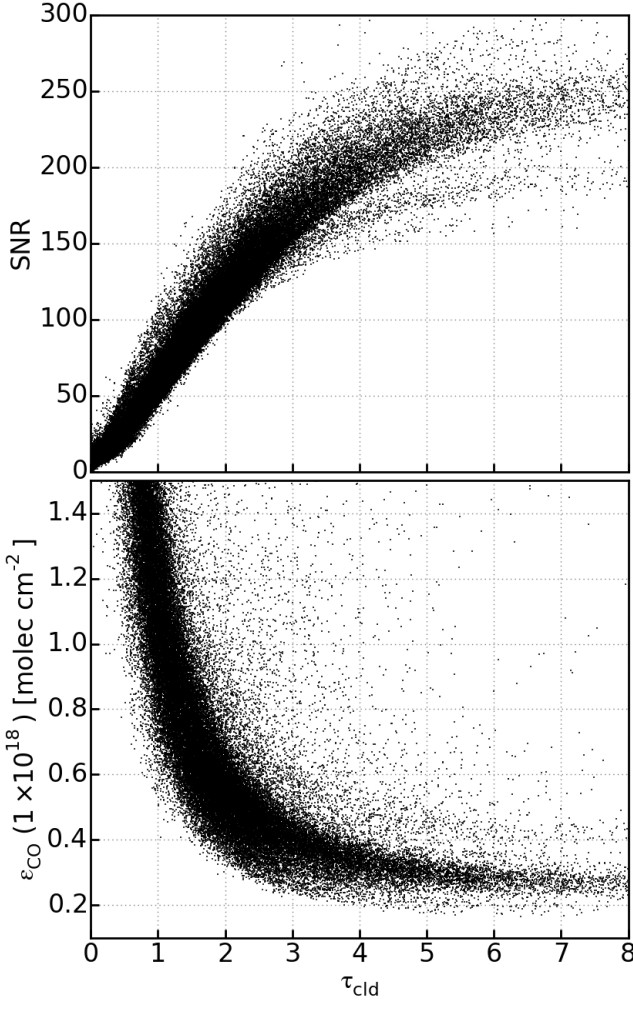

**Figure 3.** Top panel: SNR in the SCIAMACHY CO retrieval window. Lower panel: CO retrieval error $\epsilon_{CO}$ (as a function of the retrieved cloud optical thickness $\tau_{cld}$). Data is shown for the year 2003 for a latitude/longitude box [(30°S,30°W), (0°,0°)] over the ocean





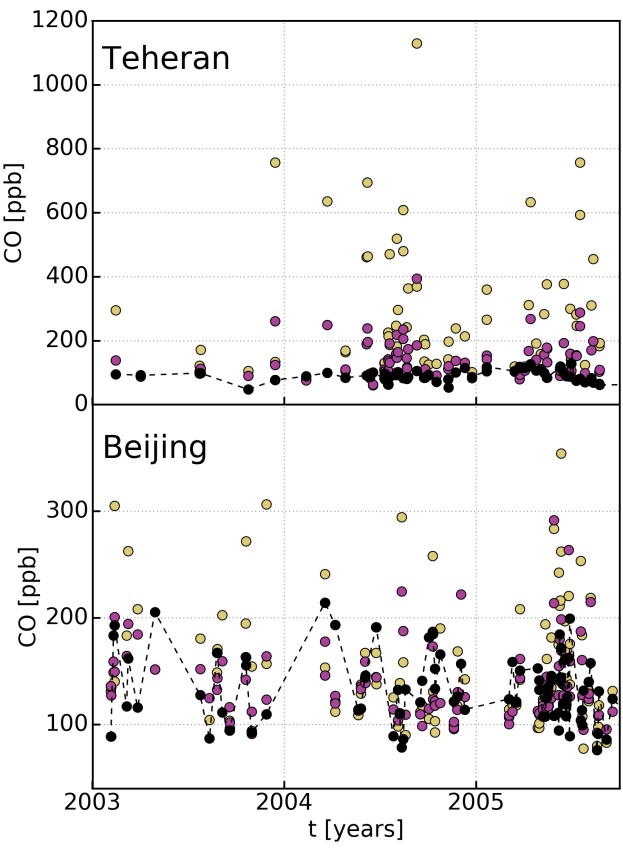

**Figure 4.** CO column mixing ratios of Teheran and Beijing measured by MOZAIC/IAGOS (yellow), MOZAIC/IAGOS with SCIAMACHY column averaging kernels applied (pink), and SCIAMACHY retrievals for optically thick high cloud conditions (black).

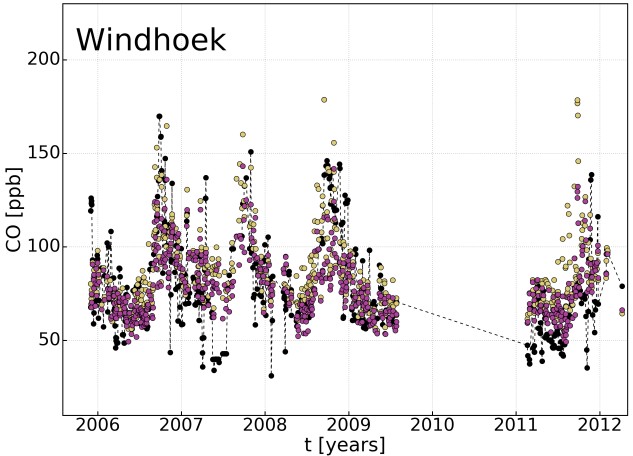

**Figure 5.** As Fig. 4, but for Windhoek.





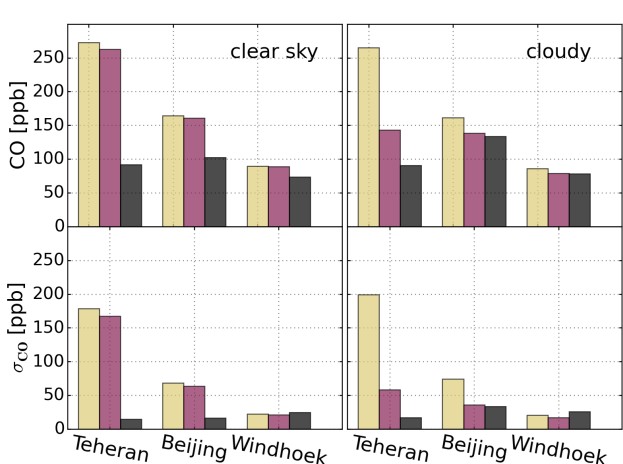

**Figure 6.** Mean CO column mixing ratio (top panel) and standard deviation (lower panel) of the time series shown in Fig. 4 and Fig. 5, SCIAMACHY (black), MOZAIC/IAGOS (yellow) and MOZAIC/IAGOS with SCIAMACHY column averaging kernels applied (pink). Left panel: clear sky SCIAMACHY retrievals and right panel: SCIAMACHY retrieval for optically thick high cloud conditions.





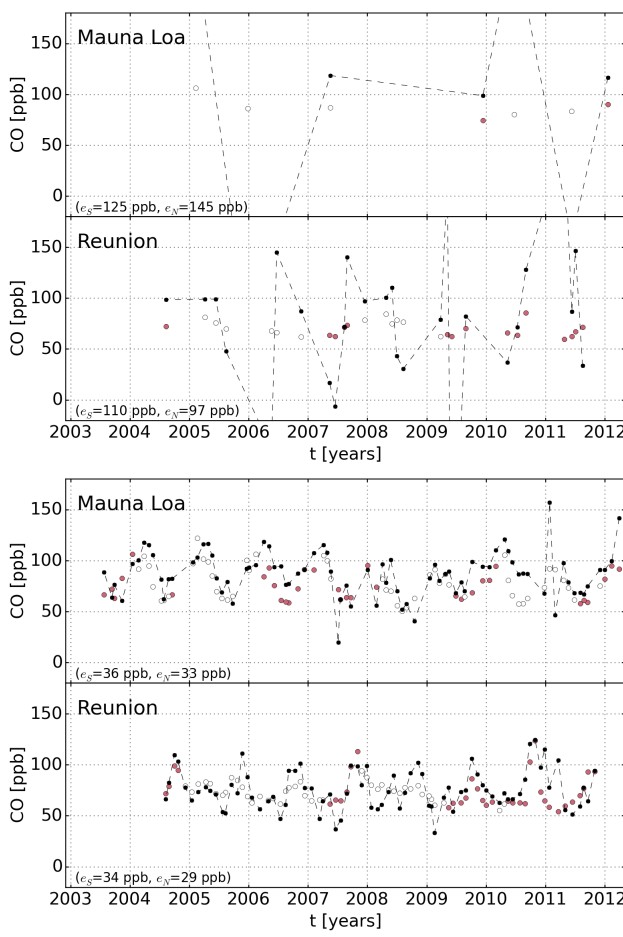

**Figure 7.** 30-day median of CO column averaged mixing ratios measured by SCIAMACHY (black) and at two NDACC-IRWG stations (pink). Upper two panels: SCIAMACHY clear-sky retrievals, lower two panels: SCIAMACHY cloudy-sky retrievals with optical thick low clouds. Open circles denote interpolated values for periods where no NDACC-IRWG measurements are available. The scatter $\epsilon_S$ and the mean retrieval noise $\epsilon_N$ for individual retrievals in the monthly bins are shown.





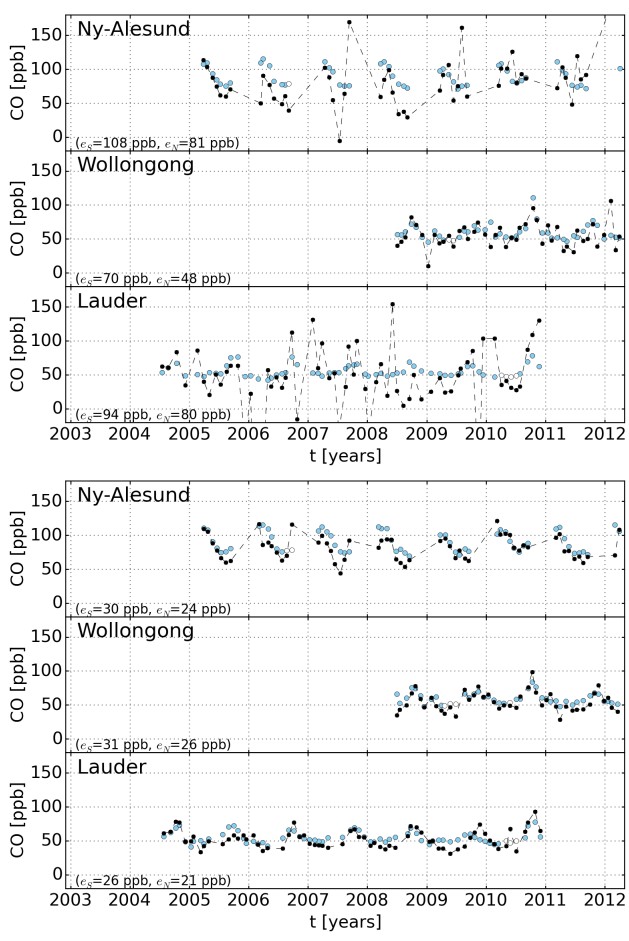

**Figure 8.** Same as Fig. 7, for TCCON measurements indicated in blue. For Wollongong and Lauder the GGG2014 release of TCCON was used, except for Ny-Ålesund data release GGG2012 is used.





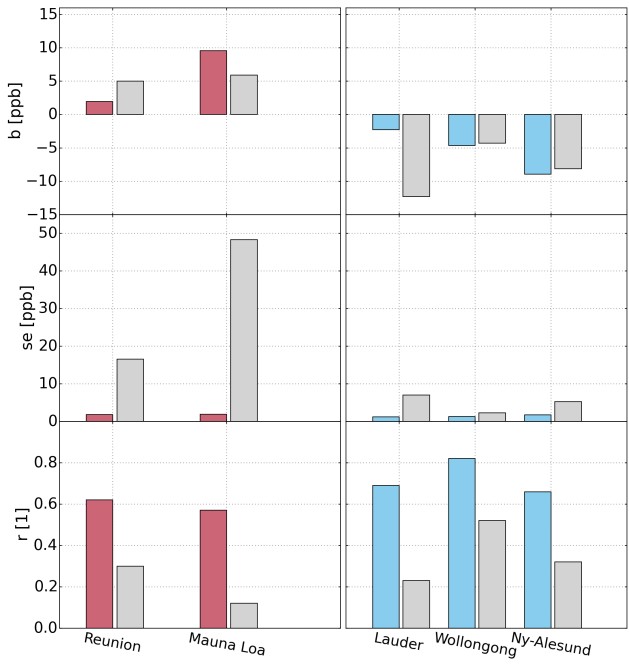

**Figure 9.** Mean bias SCIAMACHY - FTS (upper panel), standard error of the mean bias (middle panel), and Pearson correlation coefficient of SCIAMACHY with FTS (lower panel) derived from monthly medians as shown in Figs. 7 and 8. Measurements of NDACC (left) and TCCON stations (right) are compared with clear-sky (grey) and cloudy (coloured) SCIAMACHY retrievals.

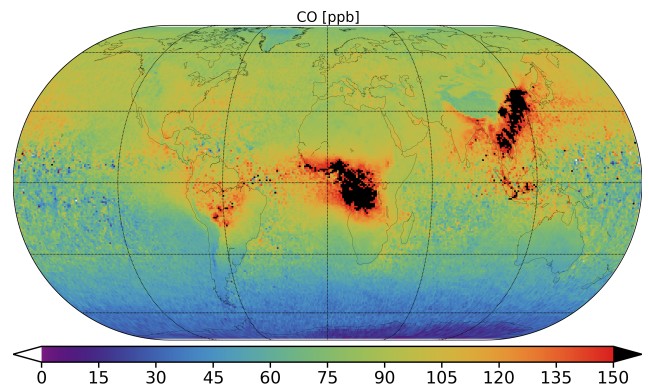

**Figure 10.** CO column averaged mixing ratios in ppb over land and ocean from clear-sky and cloudy-sky measurements for optically thick low cloud conditions. The values are averaged from January 2003 to the end of the SCIAMACHY mission in April 2012.





**Table 1.** MOZAIC/IAGOS airports used for validation. The temporal coverage with the SCIAMACHY mission is given in years.

| number | name | latitude | longitude | alt | MOZAIC/IAGOS |
|---|---|---|---|---|---|
| 1 | Beijing | 40.38° | 115.22° | 0.04 km | 2003–2005 |
| 2 | Tehran | 35.98° | 50.24° | 1.01 km | 2003–2005 |
| 3 | Windhoek | −21.43° | 17.34° | 1.72 km | 2005–2012 |

**Table 2.** Same as Table 1, but for NDACC and TCCON sites.

| number | name | latitude | longitude | alt | NDACC | TCCON |
|---|---|---|---|---|---|---|
| 1 | Ny-Älesund | 78.92° | 11.92° | 0.02 km | – | 2005–2011 |
| 2 | Mauna Loa | 19.54° | −155.57° | 3.40 km | 2003–2012 | – |
| 3 | Reunion | −20.90° | 55.49° | 0.09 km | 2004–2011 | – |
| 4 | Wollongong | −34.41° | 150.88° | 0.03 km | – | 2008–2012 |
| 5 | Lauder | −45.05° | 169.68° | 0.37 km | – | 2004–2010 |