# Peer review of "Carbon monoxide column retrieval for clear-sky and cloudy atmospheres: a full-mission data set from SCIAMACHY 2.3 $\mu$ m reflectance measurements"

_Atmospheric Measurement Techniques, 2016_

## Referee Comment (RC1) · Anonymous Referee #1 · 30 Jan 2017

This manuscript discusses carbon monoxide column retrievals from the SCIAMACHY instrument under cloudy conditions complementing previous work that was limited to clear-sky observations over land. The analysis includes comparisons with MOZAIC/IAGOS airborne measurements and with measurements of the NDACC and TCCON network at coastal sites.

The manuscript falls into the scope of AMT, is well written, and I recommend publication after the following comments have been addressed.

[Figure]

**General Comments**

Extension of the spatial coverage (especially over the ocean) by analyzing cloudy scenes is desirable. However, for thick clouds the sensitivity to CO below the cloud is very limited. As shown in the manuscript this does not hamper validation, because the corresponding null-space error can be taken into account mathematically by applying the averaging kernels. One could argue that this makes good validation results even easier to achieve, because potentially highly variable CO partial columns below clouds are essentially substituted by apriori partial columns for both the satellite data and the validation data set in the comparison. This makes physical interpretation of the retrieval results in cases of high clouds, in particular above CO source regions, difficult, because the retrieved parameter is rather the CO partial column above the cloud (extended by the apriori below the cloud) than the CO total column and the most interesting part may be hidden below the cloud. In this sense, validation and physical significance can potentially be two different stories. Please elaborate a little more on this issue in the manuscript.

Therefore, I assume that retrievals for (too) high clouds are filtered out in the post-processing. Please describe in more detail what the corresponding threshold for the presented SCIAMACHY data set is (e.g., in Figure 10). I would propose to omit scenes with cloud heights larger than about 2 or 3 km: This would still allow retrievals above low-étage clouds like stratocumulus (Sc) over the ocean.

**Specific Comments**

Page 8, Lines 13-14 and Figures 4-6: Why are cloudy conditions defined as high clouds here (condition (3) in Sect. 2.3)? The comparison with airborne measurements is mainly above CO source regions (Teheran, Beijing). This is an example of the interpretation problems described in the "General Comments". Figure 6 confirms that a lot of CO is hidden below the clouds (comparison of yellow and pink bars). I would prefer

to use condition (2) here or an alternative condition (2b) with $h_{cld} < 2$ or $3$ km.

Page 9, Lines 23-24: How about applying TCCON/NDACC averaging kernels to profile information from the TM5 CO model?

Page 10, Line 16: The given range of $h_{cld}$ corresponds to condition (3) in Sect. 2.3, not (2). Which condition is actually used, (2) or (3)? I would recommend to do the analysis for low clouds only (Condition (2) or (2b)).

Page 10, last paragraph and Figure 10: 1) Which conditions for the full-mission SCIA-MACHY cloudy-sky measurements are used? As described above, I would propose to omit scenes with cloud heights larger than about 2 or 3 km.
2) You describe CO outflow over the ocean. As the retrievals over the ocean are from cloudy-sky measurements only and the retrievals over land are a mixture from clear-sky and cloudy-sky measurements, are there any indications of land-sea-biases due to cloud shielding? This is hard to see in Figure 10, because the color-scale is often saturated at the mentioned source regions.
3) It would also be interesting to show a global map illustrating the fraction of cloudy-sky measurements. Are the source regions dominated by clear-sky or cloudy-sky measurements?

Figure 8: Why is TCCON data release GGG2012 (instead of GGG2014) used for Ny-Alesund?

**Technical Corrections**

Page 8, Line 30: Typo? "Thus" → "This"

Page 9, Line 26: Please add ":": "...is most reliable: Ny-Alesund, ..."

---

## Author Comment (AC1) · 27 Mar 2017

We would like to thank reviewer 1 and 2 for the constructive comments that aided us to improve our manuscript. In this post we provide our replies to the reviewer's comments. We provide a revised version of the manuscript, in which all changes are highlighted. Revised and added text is provided in blue. In our replies to the comment we provide line numbers, page numbers and figure numbers of the old version of the manuscript (please see the pdf-file in the supplement

Please also note the supplement to this comment:
http://www.atmos-meas-tech-discuss.net/amt-2016-355/amt-2016-355-AC1-supplement.pdf

---

## Author Response (AR1)

**Final author comments on the manuscript amt-2016-355, reviewer 1**

We would like to thank reviewer 1 for the constructive comments that aided us to improve our manuscript. In this document we provide our replies to the reviewer's comments. The original comments made by the reviewer are numbered and typeset in italic and bold face font. Following every comment we give our reply. We provide a new version of the manuscript but in our replies to the comments we provide line numbers, page numbers and figure numbers referring to the original version of the manuscript, if not stated differently.

1. Extension of the spatial coverage (especially over the ocean) by analyzing cloudy scenes is desirable. However, for thick clouds the sensitivity to CO below the cloud is very limited. As shown in the manuscript this does not hamper validation, because the corresponding nullspace error can be taken into account mathematically by applying the averaging kernels. One could argue that this makes good validation results even easier to achieve, because potentially highly variable CO partial columns below clouds are essentially substituted by apriori partial columns for both the satellite data and the validation data set in the comparison. This makes physical interpretation of the retrieval results in cases of high clouds, in particular above CO source regions, difficult, because the retrieved parameter is rather the CO partial column above the cloud (extended by the apriori below the cloud) than the CO total column and the most interesting part may be hidden below the cloud. In this sense, validation and physical significance can potentially be two different stories. Please elaborate a little more on this issue in the manuscript. Therefore, I assume that retrievals for (too) high clouds are filtered out in the post- processing. Please describe in more detail what the corresponding threshold for the presented SCIAMACHY data set is (e.g., in Figure 10). I would propose to omit scenes with cloud heights larger than about 2 or 3 km: This would still allow retrievals above low-etage clouds like stratocumulus (Sc) over the ocean.

**adjusted** The reviewer argues that the evaluation of the CO bias is distorted in case of high clouds because both the satellite data and the validation data set are substitute by the same apriori partial columns in the comparison and so this effectively reduces relative errors on the column. To refute this argument, we'd like to outline an important feature of the profile scaling approach used in our study. Here, the retrieval approach scales a reference profile to fit a forward model to the measurement. In case of a cloudy atmosphere, the scaling parameter relies on the CO measurement sensitivity above the cloud but adjusts the CO concentrations at all altitudes due to the scaling. If the scaling parameter is retrieved with a certain error, this effects the CO concentrations at all altitudes in the same manner.

So for a cloudy-sky retrieval, a measurement or forward model error affects the CO sub-column above the cloud in the same manner as it does for the total column. Therefore, the physical interpretation of the relative biases is the same for clear-sky and cloudy retrieval.

The fact that the CO bias for the site Windhoek in Fig. 7 is similar for clear-sky and cloudy retrievals, indicates a similar retrieval accuracy for CO at altitudes the measurement is sensitive to. In this sense, we are convinced that the comparison does not favor a validation of cloudy retrieval over clear-sky retrieval.

To emphasize this aspect, we added the following sentences at p6,124:

"For the interpretation of errors, it is important to note that for the scaling of a reference profile, the interpretation of relative biases is very similar for clear-sky and cloudy retrieval. In both cases, it indicates the relative error to the CO concentrations at all altitudes the retrieval is sensitive to."

Moreover, we add at p9,114:

" Consequently, the fact that clear-sky and cloudy retrievals result in the same bias indicates that the accuracy for CO is comparable at altitudes the measurements are sensitive to. "

and at p4, l23:

"Hence, when considering  $c_{\text{eff}}$  as an estimate of the true column, data interpretation does not rely on the use of the total column averaging kernel but the null-space error  $e_{\text{null}}$  becomes part of the error budget. When  $c_{\text{eff}}$  is seen as an effective column with its vertical sensitivity described by the column averaging kernel, data analysis focuses more on the information that can be inferred from the measurements and reduces the impact of the a priori choice of the reference profile. This requires a proper use of the total column averaging kernel as shown in Eq. (4) and the null-space error does not contribute to the error budget (see also Wassmann et al. (2015)). "

For validation with TCCON and NDACC in Fig 9 and to illustrate the global CO distribution in Fig. 10, it is important that the CO column have similar vertical sensitivity and can be interpreted as an estimate of the true total column. Therefore, for this figures, we use only data with a retrieved cloud top height i 1.5 km. However, based on the MOZAIC comparison, we are convinced that also CO retrievals with

higher clouds are very useful when exploiting the data set, e.g. due to the assimilation of the CO product in a chemical transport model. Therefore we are reluctant to filter the data a posteriori as suggested by the reviewer.

2. Page 8, Lines 13-14 and Figures 4-6: Why are cloudy conditions defined as high clouds here (condition (3) in Sect. 2.3)? The comparison with airborne measurements is mainly above CO source regions (Teheran, Beijing). This is an example of the interpretation problems described in the General Comments. Figure 6 confirms that a lot of CO is hidden below the clouds (comparison of yellow and pink bars). I would prefer to use condition (2) here or an alternative condition (2b) with held j 2 or 3 km.

changed De Laat et al. 2012 showed that the validation of SCIAMACHY clear sky measurements at these sites close to CO source regions suffer from representation errors, i.e. that due to the large pixel size of SCIAMACHY the aircraft measurements cannot agree with the satellite observations because of probing different CO air masses. The same holds for low clouds. However, for high clouds, the horizontal distribution of CO above the cloud is more homogenous and so representation errors are less relevant. As explained in our reply to the reviewer's general comment, the comparison between satellite and aircraft measurements are not distorted because of the profile scaling approach, and so the good agreement for SCIAMACHY observations with high clouds, support the interpretation of de Laat et al., 2012 and in addition gives confidence in the SCIAMACHY cloudy data product. To clarify this, we change the discussion in the manuscript (p8,l26) from

"The time series in Fig. 5 show strong outliers in the MOZAIC/IAGOS CO column densities, which are not found by SCIAMACHY. de Laat et al. (2012) explained this with strong local pollution at the airports, which affects the airborne measurements but are not seen by the satellite due to the coarse spatial resolution. Figure 7 confirms this reasoning. Applying the column averaging kernels of the cloudy-sky retrievals to MOZAIC/IAGOS measurements, the enhanced CO column values are strongly reduced for both Teheran and Beijing and brings the aircraft and satellite observations into much better agreement. Thus confirms that the column averaging kernel of the cloudy-sky retrieval describes well the cloud shielding of the lower atmosphere (see Fig. 2) where the local pollution is primarily located. " to

"de Laat et al. (2012) explained the larger errors for clear-sky observations with strong local pollution at the airports, which affects the airborne measurements but are not seen by the satellite due to the coarse spatial resolution. This is concert with the extremely large values of the MOZAIC/IAGOS CO column densities time series in Fig. 4. Hence, we expect a better agreement, when we consider SCIAMACHY retrievals for high clouds because the atmospheric shielding of the spatial heterogeneity of CO. Figs. 4 and 2 confirms this. Applying the column averaging kernels of the cloudy-sky retrievals to MOZAIC/IAGOS measurements significantly improve the comparison with SCIAMACHY retrievals and so supports the error interpretation by (de Laat et al., 2012) but also demonstrates the data quality of SCIAMACHY cloudy-sky retrievals. "

**3. Page 9, Lines 23-24: How about applying TCCON/NDACC averaging kernels to profile information from the TM5 CO model?**

**not changed** The paragraph at p9, 123-24 is about MOZAIC/IAGOS profile measurements and their use to validate our product. These airborne measurements can be used to validate clear-sky and cloudy SCIAMACHY retrievals and so provide interesting information on the consistency of the data product and the relevance of representation errors for the validation. This is different for TCCON/NDACC measurements, which can only be used to validate clear-sky retrievals and cloudy-sky retrievals for low clouds at remote regions. Furthermore, applying averaging kernels on TM5 profiles will not help because those are used as reference profiles in our retrieval. In Borsdorff et al. (2014) we showed that the reference profile cannot be used to fill up the null-space, i.e.  $(C - A_{col})x_{TM5} = 0$ .

Page 10, Line 16: The given range of hcld corresponds to condition (3) in Sect. 2.3, not (2). Which condition is actually used, (2) or (3)? I would recommend to do the analysis for low clouds only (Condition (2) or (2b)).

corrected Correct, we use condition (2) here. We changed the sentence at (p10, 116) from

"When analyzing SCIAMACHY cloudy-sky CO retrievals with 1.5 km  $< h_{cld} < 8$  km,  $\tau_{cld} > 2$  corresponding to condition (2) ..."

"When analyzing SCIAMACHY cloudy-sky CO retrievals with  $h_{cld} < 1.5$  km,  $\tau_{cld} > 2$  corresponding to condition (2) ..."

to

5. Page 10, last paragraph and Figure 10: 1) Which conditions for the full-mission SCIA-MACHY cloudy-sky measurements are used? As described above, I would propose to omit scenes with cloud heights larger than about 2 or 3 km.

**changed** We clarified this point in the manuscript as described in our answer to comment 4 of this review (p10, l16). Furthermore, we changed the sentence at p10,26 from: "The full-mission averaged CO product is shown in Fig. 10 and indicates ...."

to

"The full-mission averaged CO product is shown in Fig.10 and is based on clear-sky and cloudy observation contaminated by optical thick low clouds ( $h_{cld} < 1.5$  km). This figure indicates ..."

6. 2) You describe CO outflow over the ocean. As the retrievals over the ocean are from cloudy-sky measurements only and the retrievals over land are a mixture from clear-sky and cloudy-sky measurements, are there any indications of land-sea-biases due to cloud shielding? This is hard to see in Figure 10, because the color-scale is often saturated at the mentioned source regions.

**changed** To consider potential error correlation with land-sea transitions, we investigated biases for the coastal site Wollongong but did not find any indication for a land-sea bias in the data. To emphasize this point, we changed the sentence (p10, l21) from:

" For Wollongong, the mean bias for clear-sky and cloudy-sky retrievals is comparable (about 4.0 ppb), but the correlation improves from about r = 0.5 to 0.8 for the cloudy retrievals."

 $\operatorname{to}$

"For Wollongong, the mean bias for clear-sky and cloudy-sky retrievals is comparable (about 4.0 ppb), and the correlation improves from about r = 0.5 to 0.8 for the cloudy retrievals. Here the majority of the cloudy measurements are observations over the oceans and so these results indicate that there is no evidence for a land-sea bias of the SCIAMACHY CO retrieval."

7. 3) It would also be interesting to show a global map illustrating the fraction of cloudy-sky measurements. Are the source regions dominated by clear-sky or cloudy-sky measurements?

 $\mathbf{added}$  We included the figure suggested by the referee. The discussion is added to the manuscript at  $\mathbf{p7,l9}$

"Figure 4 shows that much more data become available when considering cloudy-sky retrievals under optical thick low and high cloud conditions in addition to clear-sky retrievals. Between January 2003 to the end of the SCIAMACHY mission in April 2012, 70 % of all data are inferred from cloudy-sky observations (40% over the oceans and 30% over land). Over the oceans only cloudy-sky retrievals are possible but over land the fraction of clear-sky retrievals write. For example, the data coverage over the Sahara region is dominated by clear-sky retrievals while the Amazon region and Indonesia show cloudy-sky retrievals. Hence for SCIAMACHY, our CO retrieval algorithm for cloudy conditions permits the CO data product over the ocean, and in addition has the potential to improve the product over land."

8. Figure 8: Why is TCCON data release GGG2012 (instead of GGG2014) used for Ny-Alesund?

added We added this explanation to the figure description: "... for Ny-Ålesund data release GGG2012 is used because the newer release GGG2014 is not yet available for this site."

- 9. Page 8, Line 30: Typo? Thus This corrected
- 10. Page 9, Line 26: Please add :: ...is most reliable: Ny-Alesund, ... corrected

**References**

- Borsdorff, T., Hasekamp, O. P., Wassmann, A., and Landgraf, J.: Insights into Tikhonov regularization: application to trace gas column retrieval and the efficient calculation of total column averaging kernels, Atmos. Meas. Tech., 7, 523–535, doi:10.5194/amt-7-523-2014, 2014.
- de Laat, A. T. J., Dijkstra, R., Schrijver, H., Nédélec, P., and Aben, I.: Validation of six years of SCIAMACHY carbon monoxide observations using MOZAIC CO profile measurements, Atmos. Meas. Tech., 5, 2133–2142, doi:10.5194/amt-5-2133-2012, 2012.

Wassmann, A., Borsdorff, T., aan de Brugh, J. M. J., Hasekamp, O. P., Aben, I., and Landgraf, J.: The direct fitting approach for total ozone column retrievals: a sensitivity study on GOME-2/MetOp-A measurements, Atmospheric Measurement Techniques, 8, 4429–4451, doi:10.5194/amt-8-4429-2015, 2015.

**Final author comments on the manuscript amt-2016-355, Ruediger Lang**

We would like to thank Ruediger Lang for the constructive comments that aided us to improve our manuscript. In this document we provide our replies to the reviewer's comments. The original comments made by the reviewer are numbered and typeset in italic and bold face font. Following every comment we give our reply. We provide a new version of the manuscript but in our replies to the comments we provide line numbers, page numbers and figure numbers referring to the original version of the manuscript, if not stated differently.

1. Retrievals involving the knowledge of certain above-ground mean-scattering heights with negligible information retrieved from below the scattering layer require special pre-cautionary measures taken by the producer to avoid mixing of total and partial column information. The authors mention in Section 2.3 on page 6 the potential of providing complementary information due to this mixing, but do not go in any detail how this complementary information (what is meant is probably profile information) can be retrieved or is used.

changed We modified the paragraph (p6,l32)

"CO retrievals for cloudy conditions represent an interesting addition of our data product because of the different vertical sensitivity, the retrieved CO column probes different altitude ranges and so providing complementary information..." to

"CO retrievals under cloudy conditions represent an interesting addition to our clear-sky data product. Here, the retrieved column is mostly sensitive to CO above the cloud and for varying cloud height and due to the shielding effect of clouds, different altitudes are probed. For the interpretation of errors, it is important to note that for the scaling of a reference profile, the interpretation of relative biases is the same for clear-sky and cloudy retrieval. In both cases, it indicates the relative error to the CO concentrations at all altitudes the retrieval is sensitive to. Furthermore, "

2. I find the way the retrieved column is interpreted relative to the true profile in equation 4 quite misleading with respect to the formulation in the conclusion saying: Generally for clear-sky observations the null-space error is small, but for cloudy conditions it can easily exceed 30 percent. Here, clouds shield the atmosphere below and the a priori CO profile shape is used to add the lacking information. Here it is formulated as if ghost column information from the scaled a-priori profile is added to complement the lacking information of the retrieved column above the scattering layer, while in Equation 4 it looks like the profile information is over-weighted with an averaging kernel above the cloud, therefore using only (or predominantly) a priori profile information from above the scattering layer in the retrieval. What is true?

text adjusted The reviewer's point does not represent a contradiction the two mentioned statements are mathematically equivalent as described in Borsdorff et al. (2014). To clarify this, we adjusted the text at p6 line 12, referring to our previous work to keep the discussion short.

"Column averaging kernel values of one represent the ideal value for a vertical integration (see Eq. (2)) and values < 1 indicate reduced retrieval sensitivity e.g. because of atmospheric shielding by clouds and aerosols. Values > 1 are typical for a profile scaling approach. In case of an optically thick cloud the sensitivity of the measurement for CO below the cloud is lost. The inversion scales the entire vertical profile of CO based on its sensitivity to CO above the cloud. Hence, the sensitivity of the retrieved CO column with respect to the CO concentration above the cloud is enhanced (> 1) and at the same time the sensitivity for CO below the cloud is reduced (< 1). These features are clearly reflected in Fig. 2 and discussed in more detail by Borsdorff et al. (2014). "

3. In addition, the potential draw-backs of providing this mix of effective and total column information is not discussed in much detail in the paper. While this type of data-set can potentially be used relatively safely in model assimilation, where the use of the averaging kernels is an integral part of the assimilation process, it is less clear how such a data-set can or should, if at all, be used in the way it is displayed in Figure 10. This does not render the data-set useless of course. However it requires a discussion on the purpose of the data-set presented here.

**changed** We agree with the reviewer that in general the use of the column averaging kernel is essential for proper data interpretation, as it is stated in the manuscript (p6 l22-24). For validation, CO profile measurements are required, which are only available at a few sites. A corresponding validation strategy is discussed in Sec. 3.1. We also argue that for low clouds over unpolluted NDACC/TCCON sites, the retrieved columns can be considered as a reasonable estimate of the true column density of CO within

the overall uncertainty of the SCIAMACHY CO measurement. This is demonstrated in Sec. 3.2 showing validations with TCCON and NDACC measurements. To illustrated the new SCIAMACHY data product, we included these low cloud observations in Fig 11 (previous 10) and added the following sentence to the manuscript(p10,l26): "In particular, the validation with NDACC and TCCON measurements revealed that the CO column, retrieved from measurements with low, optically thick clouds, can be used as an estimate of the true column (i.e. ignoring effects of the column averaging kernel) in absent of local sources of CO. "

4. From my point of view, such a SWIR CO SCIAMACHY data-set is only complete once the results presented have been assimilated in a model, i.e. as a level-3 or 4 products, and model information has been added to the part of the profile not visible to the retrieval. From this perspective the scope of this paper would then be the presentation of a major step towards such a final level-4 data-set. I therefore suggest to add a discussion of this aspect (maybe even consider to change the title to sub-column retrievals) and consider to add an outline on how to complete the task, in a next step. E.g. by providing a final SCIA CO SWIR data-set by adding missing sub-column information for individual cloudy-sky retrievals involving a model or other means (i.e. reducing the null-space error). We should not forget that there are currently many atmospheric composition and other retrieval methods available which potentially can provide an-above-cloud-column (e.g. water vapour retrievals from the same type of instrumentation) but choose instead to provide data-gaps in order to avoid confusion and biases, even though averaging kernels are available as well for these retrievals. While overcoming instrument SN deficiencies are the means to an end it cannot be the end for the means.

**changed** We are reluctant on a statement that the data set is only completed after assimilating the retrieval results. Each data level has its value set and should be validated based on the data uncertainty of the subjacent data level. Therefore, reporting on the level-2 CO SCIAMACHY data product and its validation we consider as a valuable contribution. We are also convinced that applications of the level-2 product other than data assimilation are of scientific value and should not be excluded a priori. On the other hand, we also agree that data assimilation as e.g. considered by the European CAMS project is an important application and should be investigated in future studies. Therefore we added to the conclusions (p12,l19):

"... of the SCIAMACHY CO data product. The correct interpretation of these data requires the use of the column averaging kernel. For varying cloud height and due to the shielding of the atmosphere below the cloud, the data may be used in future work to discriminate the vertical distribution of CO in the atmosphere, e.g. by means of data assimilation. "

**5. Abstract: line 10: low bias...: Some SCIAMACHY CO retrievals apply a bias correction to the CO product, e.g. to agree with MOPITT at source free locations (Pacific) or with trusty ground stations. Is any bias correction applied prior to this comparison?**

**changed** We do not apply any a posteriori bias correction on CO. However, as shown in Borsdorff et al. (2016), Fig 4, we apply a radiometric correction to the spectral measurements of SCIAMACHY. This correction is estimated from high albedo measurements over Sahara assuming that the chemical transport model TM5 provides the true methane profile, e.g. p5,15.

"To account for radiometric errors of the SCIAMACHY measurements, we apply a temporal dependent radiometric correction derived from SCIAMACHY observations of the Sahara and reliable model prediction of methane over this area. Moreover, the measurement noise is estimated from regular eclipse measurements and both aspects are described by Borsdorff et al. (2016). "

6. Abstract: line 20: overall the cloudy-sky... Considering the improvements exposed above, would not cloudy-sky measurements be something more than simply a "valu- able addition"?

**changed** We agree and adjusted the abstract at (p1,l20) from

"Overall the cloudy-sky CO retrievals from SCIAMACHY short wave infrared measurements present a valuable addition to the clear-sky only data set."

 $\operatorname{to}$

"Overall the cloudy-sky CO retrievals from SCIAMACHY short wave infrared measurements present a major extension of the clear-sky only data set, which more than triples the amount of data and adds unique observations over the oceans."

7. Introduction: line 7: two stream radiative transfer solver.. Are two streams sufficient to accurately simulate observed radiances at top of the atmosphere? This simplification could lead to observation geometry dependent biases. Are sphericity effects taken into account?

**changed** This point is discussed in detail by Landgraf et al. (2016). For individual cases, the induced errors may reach several percent but becomes insignificant when considering regional scales. To address this aspect, we change the sentence at (p3,l7) from:

" These parameters are used in a two stream radiative transfer solver called 2S-LINTRAN (Landgraf et al., 2016) to account for the changes to light path that occur due to scattering by clouds and aerosols."

to

"These parameters are used in a two stream radiative transfer solver 2S-LINTRAN to account for changes to the light path due to light scattering by clouds and aerosols. For cloudy conditions, the CO error induced by this radiative transfer solver reaches occasionally several procent (Landgraf et al., 2016) and becomes marginal for regional measurement ensembles. The SICOR data product"

8. Section 2.1, line 23: Are the SCIAMACHY operational L1 noise values reliable? Otherwise
- although formally correct - the consideration of the measurement noise could worsen the CO estimates. Or is the CO product derived from a different SCIAMACHY L1 product?

**changed** We are not using the operational L1 noise as described by Borsdorff et al. (2016), p232: "The channel 8 SCIAMACHY measurement noise is dominated by detector dark noise, which is estimated from SCIAMACHYs daily dark state measurements taken during the orbit eclipse. "We found that this approach improves significantly our retrieval because the noise of every detector pixel is basically measured as a function of time.

A reference and a corresponding remark is added at p5 l29 as stated in reply 5: " For SCIAMACHY CO retrievals, we use the recalibrated SCIAMACHY spectra including estimates of the measurement noise as described by Borsdorff et al. (2016). "

9. Section 2.2, page 5, 1st paragraph: A short motivation for increasing or choosing this spectral window would be helpful (like for the reduction of noise. information content, type of absorbers, ice layer contamination etc...).

**changed** The selection of the retrieval window is described in detail by Borsdorff et al. (2016). To clarify this we add the sentence (p5,l5): "....This is necessary to account for the instrument degradation e.g. the considerable loss of detector pixel by radiation damage and the light scattering of the ice-layer on the detector array of SCIAMACHY's channel 8 (Borsdorff et al., 2016). "

10. Section 2.2, page 5, line18: If the used cloud model has a fixed half-width of 1.5km, the lowest possible cloud top height in the forward model is 3km. Does this not constrain the retrieval in an unnecessary way?

**changed** The retrieval is not restricted to a lowest cloud height > 3 km but the vertical profile of the cloud is truncated at the Earth surface. To clarify this we add the following sentence (p5,l19):

"In case of cloud heights < 3km the cloud profile is cut off at the Earth surface and is renormalized"

11. General: I find the term "physics-based" retrieval bit problematic. It suggests that you may do certain things somewhere else, which are not physics based. Do you? I hope not! Full-physics would be also problematic, since many effects have not been considered in the retrieval, e.g. horizontal variability inside the spatial pixels.

changed We change the term "physics-based retrieval" to "scattering retrieval" through out the manuscript.

12. Section 2.2, page 5 last paragraph: The cloud optical properties depend on the microphysical ones. Accordingly, the retrieval results depend on the assumed cloud micro- physics. Could you be more specific on the cloud microphysical model which has been used? not changed

For the retrieval of effective cloud properties as done by SICOR, the step to derive optical properties from microphysical properties is not required. For example, it is sufficient to describe the cloud by its optical depth and the asymmetry factor of the scattering phase function without analyzing the underlying droplet size distribution and refractive index. The approach is discussed in detail by Landgraf et al. (2016) and an appropriate reference is already given in the manuscript.

13. Section 2.3, page 6: After 2005, SCIAMACHY channel 8 detector suffered irreversible damage decreasing the quality of the spectra. Do the FRESCO and SICOR cloud heights also correlate that good for other years after 2005?

**changed** The year 2005 is not a specific year concerning the SCIAMACHY channel 8 detector performance, which actually suffered from a continuous degradation. Nevertheless, the SICOR cloud height retrieval is only affected marginally by this degradation. We correlated the FRESCO data with the SICOR cloud height for the year 2010 and found a similar good comparison. Therefore, we added a sentence at (p6,l5) and (p11,l17)

"Even for later years of the mission we find a similar agreement (e.g. r = 0.8, slope= 0.9, intercept=431 m for the year 2010)."

14. Section 2.3, page 6, line 14ff: In case of an optically thick cloud... It is not clear from this formulation whether scaling of the profile in a retrieval under cloudy conditions is performed differently or in the same way as in the clear sky case. When it is stated that "... using only the measurement sensitivity above the cloud", does it mean that "the atmosphere bellow the cloud is neglected and not taken into account during the fitting process" or that "the forward model, which includes a cloud layer, is practically insensitive to CO bellow the cloud and mainly fit the CO information above the cloud"? Please clarify.

**changed** The retrieval still fits a scaling of a reference profile but the forward model becomes insensitive due to the presence of a cloud. To clarify this, we changed the manuscript according to comment 2 of this reviewer.

15. Section 2.3, I am missing a discussion of the fact that you expect/want the cloud parameters to be different from the FRESCO once retrieved in the NIR in first place. Since otherwise you could simply use the latter to at least adjust your  $h_{-}$ cld. I think you should mention here the reasons why and how you expect them to be different in the SWIR. For sure one expects them to be higher in the SWIR, which is what one can see.

**changed** FRESCO and SICOR use a different way to describe clouds in an effective manner. For example, FRESCO uses a elevated Lambertian reflector of predefined albedo which partially covers the ground scene. SICOR uses a homogenous scattering layer with triangular height profile and variable optical depth. Moreover, the SCIAMACHY NIR and SWIR measurements are not necessarily collocated, which may introduce a pseudo-random error contribution. Therefore, we add the following sentence at (p6,l5):

"The remaining small difference between the retrieved cloud height and the FRESCO cloud top height may origin from the different cloud models used. For example, FRESCO simulates clouds as an elevated Lambertian surface, whereas SICOR fits the center height of the triangular height profile."

16. Section 2.3, page 6, line 27: Also for the validation.... Strictly speaking the data-use is limited. Since its limited to the profile part where the null-space is small. While the sentence is correct in principle it paints a too broad picture of data use.

**changed** We change the sentence starting from (p6,l25)

"Generally, the null-space error does not limit the data use if the averaging kernel is applied properly. For example, for data assimilation the averaging kernel provides the altitude sensitivity, which is needed to adequately adjust the atmospheric state. " to

"The null-space does not provide a problem for application like data assimilation, if the averaging kernel is applied properly. Exploiting the altitude sensitivity given by the averaging kernel, the atmospheric state of the model can be adjusted adequately by the assimilation scheme. Also for the validation of the CO data product, the null-space error does not impose principle limitation, where Eq. (5) accounts for cloud effects in our retrieval when comparing satellite measurements with independent profile soundings."

17. Section 2.3, page 6, line 32: CO retrievals for cloudy conditions... Is the complementary use of CO columns with different averaging kernels to retrieve vertical profile information demonstrated somewhere in the paper (see general remark)? In contrast, the reduction of the SN and the consequent reduction on the retrieval noise is clearly demonstrated and should function as the key-motivation to use cloudy scene retrievals.

**adjusted** In the manuscript we discuss the use of the retrieved column for cloudy atmospheres with varying cloud height to infer CO height information as an potential future application, which can be achieved by data assimilation. To make this point clear, we adjusted the text accordingly to our reply to reviewer's comment 1.

18. Section 2.3, page 7 last paragraph, last sentence: ...and in addition has the potential to improve the CO product over land. ->In cases where the true and the a-priori CO profile are similar, and where no significant deviations in CO concentration (true vs. a-priori) below the cloud layer occur.

**not changed** We disagree, the product is improved in more general terms (e.g. data coverage, lower noise error, ...). The data filtering mentioned by the reviewer is only required to validate our data product with TCCON/NDACC measurements. When profile measurements are available, the validation is not hampered by this, as discussed in Sec. 3.1. Here, the total column averaging kernel can be deployed and the validation is possible without any restriction on the profile shape and a priori knowledge.

- 19. Section 3: retrieval conditions: SN>20: This condition will change the CO retrieval densities favoring Earth location with high surface albedos. Have you notice any data gaps in dark surface locations? statistics not changed The SNR filter threshold is carefully chosen to thin out data as less as possible. Figure 3 shows that a SNR of about 20 corresponds to clear-sky measurements over the oceans and so by definition the data filter does cause some data gaps over ocean. However, its effect on land observation is minor.
- 20. Section 3: retrieval conditions number 3): The quality of the SCIA channel 8 spectra and, consequently, also the CO data record, gets worse after the end of 2005. Accordingly one could think in time variable noise thresholds to select for useful CO retrievals. Are the given noise thresholds constant over the whole mission?  $\varepsilon_{CO} \sim 1e19$  represents roughly a relative noise error of some hundreds of percent. Please justify this quality criterion.

**changed** Yes, the noise threshold is constant over the whole mission. It is a weak filter on the data to remove the largest outliers, which cannot be explained by the noise statistics. Therefore, it is not needed to adjust the threshold as a function of time. To clarify this we add the following sentence to the manuscript (p7,l21):

"These criteria represent a weak data filtering to remove outliers due to unphysical retrievals. For example, the signal-to-noise threshold corresponds to clear-sky measurements over the oceans (see Fig 3) for which a stable inversion is not possible. The retrieval would result in unphysical outliers and are thus rejected by the filter. "

21. Section 3.1, page 8, line 5: Furthermore, towards... Whereas this is an acceptable assumption for unpolluted regions, I would expect that CO concentrations increase towards the surface in polluted areas, since CO emissions happens at surface (vehicle and industrial emissions, biomass burning, etc). Maybe the assumption is backed by transport over the selected circular area of 850km?

**changed** We agree with the reviewer that for source regions the CO concentration may increase towards the surface. However, our experience is that a linear schemes as proposed by the reviewer introduces the risk to unstable extrapolation due to the uncertainty of the profile shortly after take-off. This is avoided by our method, which is also in agreement with the work of (de Laat et al., 2012) estimating the error of the approach to be a view percent of the total column.

We add the following sentence to the manuscript (p8,16):

"... which ensures the numerical stability of the extrapolation. When the CO column density is derived from the aircraft measurement in this manner its accuracy is estimated to be a few percent (de Laat et al., 2012). Numerical experiments using the lower data points of the profile to estimate the gradient in the CO concentration for linear extrapolation led to instabilities and thus this approach is not used here."

22. Section 3.1, page 8, line 9: ...chosen dynamically...: Can you provide the maximum and minimum temporal difference considered in the collocation?

**changed** We changed the sentence (p8,19) from: "The temporal collocation criterium is chosen dynamically for each individual MOZAIC/IAGOS measurement." to

"The temporal collocation criterium is chosen dynamically for each individual MOZAIC/IAGOS measurement and varies typical between 7 and 30 days."

**23. Section 3.1, page 9, line 3 and 4: Considering a collocation area higher than 2e6km2, the difference in city areas of about $\sim 1000km2$ do not a play a role explaining the CO results.**

**changed** we agree and remove the part

"Beijing with 18.6 million urban inhabitants and an urban area of about  $1400 \,\mathrm{km^2}$  has about double the size of Teheran with an urban population of about 8.8 million and an urban area of  $730 \,\mathrm{km^2}$ . Moreover the metropolitan area in Beijing is much larger than that of Teheran, indicating that the CO sources in Beijing may be spatially more extended than those in Teheran." from the manuscript (p9,l2-4):

24. Section 3.1, page 9, 1st Paragraph and Figure 6: How does the here presented reasoning explain the fact that cloudy sky retrievals compare so much better to MOZAIC measurements with averaging kernel applied than the corresponding clear-sky retrievals. Is the above cloud column more representative to the above cloud in-situ measurement? Or what is going on here. Especially given the fact that the cloudy and clear sky columns by Mozaic are not so different in first place.

changed To clarify this point, we adjusted the discussion in the document (p8,l26-33)

"The time series in Fig. 4 show strong outliers in the MOZAIC/IAGOS CO column densities, which are not found by SCIAMACHY. de Laat et al. (2012) explained this with strong local pollution at the airports, which affects the airborne measurements but are not seen by the satellite due to the coarse spatial resolution. Figure 6 confirms this reasoning. Applying the column averaging kernels of the cloudy-sky retrievals to MOZAIC/IAGOS measurements, the enhanced CO column values are strongly reduced for both Teheran and Beijing and brings the aircraft and satellite observations into much better agreement. This confirms that the column averaging kernel of the cloudy-sky retrieval describes well the cloud shielding of the lower atmosphere (see Fig. 2) where the local pollution is primarily located. ..."

" de Laat et al. (2012) explained the larger errors for clear-sky observations with strong local pollution at the airports, which affects the airborne measurements but are not seen by the satellite due to the coarse spatial resolution. This is concert with the extremely large values of the MOZAIC/IAGOS CO column densities time series in Fig. 4. Hence, we expect a better agreement, when we consider SCIAMACHY retrievals for high clouds because the atmospheric shielding of the spatial heterogeneity of CO. Figs. 4 and 2 confirms this. Applying the column averaging kernels of the cloudy-sky retrievals to MOZAIC/IAGOS measurements significantly improve the comparison with SCIAMACHY retrievals and so supports the error interpretation by (de Laat et al., 2012) but also demonstrates the data quality of SCIAMACHY cloudy-sky retrievals.

Another interesting feature of our comparison is the better agreement for Beijing than for Teheran, where at the same time the SCIAMACHY ...."

**25. Section 3.1, page 9, Last Paragraph: An additional reason for the good agreement is also the better SNR at Windhoek due to higher surface reflectivity**

**changed** We agree and add the following text at (p9,115):

"We complete our SCIAMACHY CO validation that is based on MOZAIC/IAGOS aircraft measurements by analyzing observations from the airport at Windhoek. Figure 5 shows good agreement between SCIA-MACHY and MOZAIC/IAGOS measurements with a clear seasonality. Here, the spatial CO distribution is less affected by local sources and so representation errors are less relevant. Moreover, the high surface albedo ensures clear-sky retrievals with high precision."

- 26. Section 3.2, page 9, line 23 and 24: Because of lack of... Is the collocation area in this case also a circle of 850km radius? changed Yes, we use the same criteria as for MOZAIC as already mentioned in the manuscript (p9,28). To emphasize this, we change this sentence from: "Using the same criteria as in Sec. 3.1 ..." to "Using the same criteria (i.e. collocation radius and quality filtering) as in Sec. 3.1 ..."
- 27. How does the criterion of 30-day window compare with the dynamically chosen time window allowing an error of the mean of < 1e17?

**changed** For clear-sky retrievals, this relation depends on the retrieval precision and so mainly on the surface albedo. Thus, for some NDACC/TCCON stations the dynamical averaging corresponds to a 7-day to 30-day averaging but for others it would be even averaging over years. We addressed this point in our answer to comment 22 of the reviewer and changed the manuscript accordingly.

28. Section 3.2, page 10, line 6: >17 ppb: According the the Mauna Loa and Reunion plots, the  $\varepsilon_S$  amounts to values larger than 100 ppb. How is this number here then to be interpreted according to the plots?

**not changed** The reviewer refers here to the scatter of individual retrievals (which can be > 100ppb for low albedo sites) and the standard error of the mean bias averaged over 10 years. These are different diagnostic tools and cannot be compared one-by-one.

29. Section 3 generally: Wouldnt it be an option the application of the AK to TM5 profiles? Of course, the null space error would remain.

**not changed** No. The TM5 profiles is used as reference profile to be scaled by the inversion. Borsdorff et al. (2016) (Eq. (26)) showed that in this case  $Ax_{TM5} = Cx_{TM5}$  and so the TM5 profile cannot be used to add null-space contributions.

**30. Conclusion, page 11, line 31: on biases: Are these biases shown in Fig. 6?**

**changed** We change the statement at (p11,l29) form:

"Direct comparison of the MOZAIC/IAGOS CO columns estimated at Beijing, Teheran, and Windhoek with collocated SCIAMACHY clear-sky CO retrievals of the CO column showed a small bias of 0.5 - 9.5 ppb, in agreement with previous studies. However for cloudy SCIAMACHY observations, the bias exceeds 120.0 ppb for Teheran and 30.0 ppb for Beijing."

 $\operatorname{to}$

"Direct comparison of the MOZAIC/IAGOS CO columns estimated at Beijing, Teheran, and Windhoek with collocated SCIAMACHY cloudy-sky retrievals of the CO column show a small bias of 0.5 - 9.5 ppb. However, for clear-sky SCIAMACHY observations, the bias exceeds 120.0 ppb for Teheran and 30.0 ppb for Beijing, which is in agreement with previous studies (Borsdorff et al., 2016, de Laat et al., 2012)."

31. Figure 4, Caption: If clouds are thick and high enough, the cases compared here represent basically the CO column from ToA down to the upper troposphere. Considering that the MOZAIC/IAGOS CO profiles are extended to ToA using TM5 data, in how far do the comparison shown here for thick high cloud cases - and after applying the SCIA AK really refer to a comparison between SCIA vs. MOZAIC/IAGOS and not SCIA vs TM5?

**changed** We agree that the validation for high and optically thick clouds with MOZAIC/IAGOS profiles is subject to an additional uncertainty due to the flight height of the aircraft. As described in the manuscript (p8,l3) we only select airborne profiles, which at least reach an altitude of 9 km but overall with mean maximum altitude of 11 km. On the other hand, we select cloudy retrievals with an cloud altitude lower than 8 km and so there is a clear overlapping altitude range between the aircraft measurements and the sensitivity of the retrieval. To stress this point, we changed the following paragraph (p7,l29):

"In this study, we consider the vertical profile measurements at three cities: Teheran and Beijing, which are known to be affected by strong local pollution events (de Laat et al., 2012), and Windhoek, which is situated in an area with higher surface albedo and thus high radiometric precision of SCIAMACHY clear-sky measurements. Moreover, at Windhoek the atmospheric CO abundance is only slightly influenced by local pollution resulting in spatially homogeneous CO fields. Table 1 provides more details on these sites. From the MOZAIC/IAGOS data set, we select only those profiles that cover at least an altitude range up to 9 km with data gaps j 1 km." to

"In this study, we consider the vertical profile measurements at three cities: Teheran and Beijing, which are known to be affected by strong local pollution events (de Laat et al., 2012), and Windhoek where the atmospheric CO abundance is only slightly influenced by local emissons resulting in spatially homogeneous CO fields. The Windhoek measurement site is situated in an area with higher surface albedo and thus high radiometric precision of SCIAMACHY clear-sky measurements. Table 1 provides more details on these sites. From the MOZAIC/IAGOS data set, we select only those profiles that reach at least an altitude of 9 km with data gaps ; 1 km. The mean maximal altitude of the selected profiles is 11 km, which overall indicates a sufficient overlap between the aircraft measurements and the sensitivity of the retrieval with cloud heights < 8 km. "

**32. Figure 10: To which extend is this plot dominated by the data of the "good SCIA SWIR years" 2003, 2004, 2005? Is the CO product displayed here considering only data from 2006 up to 2012 maintain the smoothness and the coverage properties of the given plot?**

**not changed** There is not a specific time range of "good SCIAMACHY years" because the detector degrades continuously with time. Figure. 7 and Fig. 8 show the time series of NDACC/TCCON validations for cloudy SCIAMACHY observation covering the full range of the mission with a good precision over the entire time range. Here, early SCIAMACHY years do not dominate the validation. So, we are reluctant to speak here about 'good' and 'bad' SCIAMACHY years. Particular, the year 2005, mentioned by the reviewer also in previous comments, can not be confirmed as any break point of the data quality.

**33. Abstract: line 9: This improves... -> The situation improves...**

**corrected**

**34. Section 2.1 line 7: $C = (1, ..., 1)^T - > I$ would suggest to describe it in the form of: the transpose of the column vector or similar.**

**changed** After reflecting this formulation, we agree that it does not make to much sense to define here C explicitly as a row or column vector. To become independent on the specific representation of vectors, we adjusted the text as follows: " $c = \vec{C}^T \vec{\rho}_{ref}$ . Here, T indicates the transposed of the vector  $\vec{C}$  with all

elements  $C_i = 1$ . So Eq. (2) approximates the vertical integration assuming that the entries of the CO profiles are given in subcolumns.

Equation (1) is inverted with respect to state vector  $\vec{x}$  with the solution  $\vec{x}_{ret}$  using the least squares fitting approach, where we apply the Gauss-Newton algorithm to account for the non-linearity of the forward model. "

Mathematically the suggested formulation is equivalent to the formulation in the manuscript. We see no benefit in a reformulation.

35. Section 2.1, line 12: Define gain matrix G for completeness and readability (all other quantities have been defined).

**modified** The gain matrix is defined and discussed in detail in Borsdorff et al. (2014). Repeating the definition here would mean to increase the math in the manuscript substantially, without increasing the readability of the text, to our opinion. Therefore, we suggest to add the following sentence to the manuscript (p4, l26):

"Here  $\mathbf{G}$  is the gain matrix of the inversion abd its definition is given by Borsdorff et al. (2014)."

- 36. Section 2.1 line19: an -> a corrected
- 37. p.9, line 1: "...a larger representation errors..." corrected
- 38. Section 2.2, page 5, line 9: clouds -> aerosol and clouds corrected
- 39. Section 2.2, page 5 line 10: Moreover? I think this is the purpose in first place... corrected Moreover is removed.
- 40. Section 2.2, page 5, line 16 and 17: To establish .... This sentence should probably go in the previous paragraph, since it is relevant for the retrieval of  $h_{cld}$  and  $\tau_{cld}$ . But I guess the CH4 profile is also fixed for the "physics-based" retrieval. Is it?

**not changed** The paragraph describes the scattering (so physics-based) retrieval, where cloud information is inferred from CH4 absorptions. Therefore, CH4 is fixed to the TM5 profile. So, this sentence is correct and we think well placed.

41. Section 3.1, page.8, line 30: "Thus confirms that...";

**corrected**

- 42. Section 3.2, page 9, line 20: air column -> dray-air column? corrected
- 43. Figure 2, Caption: cloud at... -> cloud center height? cloud top height? Please specify. corrected It is the center height, corrected in the manuscript.

**Carbon monoxide column retrieval for clear-sky and cloudy atmospheres: a full-mission data set from SCIAMACHY 2.3 $\mu{\rm m}$ reflectance measurements**

Tobias Borsdorff1, Joost aan de Brugh1, Haili Hu1, Philippe Nédélec2, Ilse Aben1, and Jochen Landgraf1

[revised manuscript text omitted]

---

## Author Response (AR2)

**Final author comments on the manuscript amt-2016-355, editor**

We would like to thank the editor Lok Lamsal for the comments and corrections that further improve our manuscript. In this document we provide our replies to the editor's comments. The original comments made by the editor are numbered and typeset in italic and bold face font. Following every comment we give our reply. We provide a new version of the manuscript but in our replies to the comments we provide line numbers, page numbers and figure numbers referring to the original version of the manuscript, if not stated differently.

1. *Bullet 2, response for Reviewer 1: Missing of after because in ..for high clouds because the atmospheric shielding..* corrected

2. *Bullet 2, response for Reviewer 1: Revise this statement: Applying the column averaging kernels of the cloudy-sky retrievals to MOZAIC/IAGOS measurements significantly improve the comparison with SCIAMACHY retrievals and so supports the error interpretation by (de Laat et al., 2012) but also demonstrates the data quality of SCIAMACHY cloudy-sky retrievals. Suggestion: Applying the column averaging kernels of the cloudy-sky retrievals to MOZAIC/IAGOS measurements significantly improves the comparison with SCIAMACHY retrievals, supports the error interpretation by de Laat et al. (2012), and demonstrates the data quality of SCIAMACHYcloudy-sky retrievals.* adjusted we changed the sentence as suggested by the editor.

3. *Bullet 7, response for Reviewer 1: Change optical to optically in ..considering cloudy-sky retrievals under optical thick low and high cloud conditions..* corrected

4. *Bullet 7, response for Reviewer 1: There might be unnecessary be in ..Indonesia show cloudy-sky retrievals be in a majority ..* corrected

5. *Bullet 3, response for Reviewer 2: Please change absent to absence in ..true column (i.e. ignoring effects of the column averaging kernel) in absent of local sources of CO. "* corrected

6. *Bullet 4, response for Reviewer 2: Please revise this statement for clarity. For varying cloud height and due to the shielding of the atmosphere below the cloud, the data may be used in future work to discriminate the vertical distribution of CO in the atmosphere, e.g. by means of data assimilation. "* adjusted We changed the fragment at p12,l19 from "For varying cloud height and due to the shielding of the atmosphere below the cloud, the data may be used in future work to discriminate the vertical distribution of CO in the atmosphere, e.g. by means of data assimilation."
to
" In future, the data may be used to discriminate the vertical distribution of CO in the atmosphere, e.g. by means of data assimilation. For this, retrievals under optically thick clouds with varying cloud heights are essential because the shielding of the atmosphere below the cloud reveals information about CO in different altitudes. "

7. *Bullet 5, response for Reviewer 2: Please consider replacing temporal dependent radiometric to either time-dependent radiometric or temporal dependence of radiometric.* adjusted we changed "temporal dependent" to "time-dependent'

8. *Bullet 7, response for Reviewer 2: I suggest making changes in ..occasionally several procent (Landgraf et al., 2016) and becomes marginal for .. as ..occasionally several percent (Landgraf et al., 2016), but becomes marginal for..* corrected

9. *Bullet 12, response for Reviewer 2: I am fine with your response, but appropriate statements are necessary in the manuscript to address reviewers comment.* adjusted We add the following sentence at p5, l27: " For the retrieval of effective cloud properties it is sufficient to describe the cloud by its optical depth and the asymmetry factor of the scattering phase function without analyzing the underlying droplet size distribution and refractive index as discussed in detail by Landgraf et al. (2016). "

10. *Bullet 16, response for Reviewer 2: lead to may better replace provide in "The null-space does not provide a problem for application... Please check if you meant principal in ..does not impose principle limitation...* corrected We changed "provide" by "let to" and "principle" py "principal"

11. *Bullet 19, response for Reviewer 2: It looks like this comment is not addressed in the manuscript, although you have a response for the reviewers comment.* adjusted We add at p7, l20 "Hence, by definition the data filter does cause some data gaps over ocean. However, its effect on land observation is minor"

12. ***Bullet 24, response for Reviewer 2: Please replace found by observed in ..are not found by SCIAMACHY.*** **already removed** That sentence was already removed as response to the comment of the reviewer.

13. ***Page 1, line 19 in abstract: Please replace add by adds.*** **corrected**

14. ***Page 5, line 22: of the Sahara =¿ over the Sahara.*** **corrected**

**References**

[revised manuscript text omitted]